# Nitrate assimilation and regeneration in the Barents Sea: insights from nitrate isotopes

Robyn E. Tuerena[1†*], Joanne Hopkins[2], Raja S. Ganeshram[1], Louisa Norman[3], Camille de la Vega[3], Rachel Jeffreys[3] and Claire Mahaffey[3]

[1]School of GeoSciences, University of Edinburgh, James Hutton Rd, Edinburgh, EH9 3FE, UK.
[2]National Oceanography Centre, 6 Brownlow Street, Liverpool, L3 5DA, UK.
[3]School of Environmental Sciences, University of Liverpool, 4 Brownlow St, Liverpool, L69 3GP, UK.
[†]Now at: Scottish Association for Marine Science, Oban, Argyll, UK. PA37 1QA. Oban, Argyll, UK. PA37 1QA.

Correspondence to: Robyn E. Tuerena (robyn.tuerena@sams.ac.uk)

**Abstract.** While the entire Arctic Ocean is warming rapidly, the Barents Sea in particular is experiencing significant warming and sea ice retreat. An increase in ocean heat transport from the Atlantic is causing the Barents Sea to be transformed from a cold, salinity stratified system into a warmer, less-stratified Atlantic-dominated climate regime. Productivity in the Barents Sea shelf is fuelled by waters of Atlantic origin (AW), which are ultimately exported to the Arctic basin. The consequences of this current regime shift on the nutrient characteristics of the Barents Sea are poorly defined. Here we use the stable isotopic ratios of nitrate ($\delta^{15}$N-NO3, $\delta^{18}$O-NO3), to determine the uptake and modification of AW nutrients in the Barents Sea. In summer months, phytoplankton consume nitrate, surface waters become nitrate depleted, and particulate nitrogen ($\delta^{15}$N-PN) reflects the AW nitrate source. The ammonification of organic matter in shallow sediments resupplies N to the water column and replenishes the nitrate inventory for the following season. Low $\delta^{18}$O-NO3 in the northern Barents Sea reveals that the nitrate in lower temperature Arctic Waters is >80% regenerated through seasonal nitrification. During on shelf nutrient uptake and regeneration, there is no significant change to $\delta^{15}$N-NO3 or N*, suggesting benthic denitrification does not impart an isotopic imprint on pelagic nitrate. Our results demonstrate that the Barents Sea is distinct from other Arctic shelves, where benthic denitrification enriches $\delta^{15}$N-NO3 and decreases N*. As nutrients are efficiently recycled in the Barents Sea and there is no significant loss of N through benthic denitrification, changes to Barents Sea productivity are unlikely to alter N availability on shelf or, the magnitude of N advected to the central Arctic basin. However, we suggest that the AW nutrient source ultimately determines Barents Sea productivity and changes to AW delivery have the potential to alter Barents Sea primary production and subsequent nutrient supply to the central Arctic Ocean.

## 1 Introduction

The Arctic Ocean is warming (Huang et al., 2017) and experiencing sea ice loss (Notz and Stroeve, 2016), and freshening (Coupel et al., 2015) as a direct response to climate change. It is an enclosed basin filled with waters from the Atlantic and Pacific Oceans, which provide varying concentrations of nutrients (Torres-Valdes et al., 2013). In turn, these nutrient supply pathways influence the distribution and extent of primary production throughout the Arctic Ocean (Lewis et al., 2020). Approximately 50% of the Arctic Ocean is made up of productive shelves that support large fisheries and diverse habitats (Dalpadado et al., 2014;Friedland and Todd, 2012). As the Arctic continues to warm and more sea ice is lost, phytoplankton growth will become less limited by light availability. Instead, nutrient availability, principally nitrate (Codispoti et al., 2013), may become the primary control on phytoplankton growth (Arrigo and van Dijken, 2015;Lewis et al., 2020). Further insight is required into how nitrate is supplied to Arctic shelves, the nutrient cycling processes that occur in-situ and their sensitivity to climate change. A further understanding of these processes will help to inform on future changes to Arctic primary production and food web dynamics (de la Vega et al., 2020).

Atlantic Water (AW) is supplied to the Arctic via Fram Strait and the Barents Sea Opening and fills most of the deep basins of the Arctic. It supplies nutrients to the Eurasian shelves with nitrate and phosphate concentrations close to Redfield (15-16N:1P), and low concentrations of silicate, which can limit the extent of diatom growth (Hatun et al., 2017). AW is a mixture of nutrient-rich North Atlantic subpolar and nutrient-poor subtropical origin water advected into the Norwegian Sea. Over the last two decades there has been a 7% and 20% decrease in nitrate and silicate concentrations respectively in the Barents Sea (Rey, 2012). This has been driven by shallower winter mixing in the subpolar gyre, coupled with weakening and westward retraction of the gyre which has increased the proportion of subtropical origin water entering the Norwegian Sea (Rey, 2012;Hatun et al., 2017).

As warm and saline AW inflow water is transported across the Barents Sea, it is modified by atmospheric cooling and is mixed with cold, fresh Arctic origin water (ArW) and the Norwegian Coastal Current (NCC) (Figure 1b). ArW found across the northern Barents Sea comprises fresh Arctic river runoff, sea ice melt and precipitation, and contains the remnants of the winter mixed layer (Rudels et al., 1996). Less dense ArW isolates the sea surface and ice cover from warm AW below (Lind et al., 2016) and during the summer is capped by a well-mixed surface layer of fresh melt water (Polar Surface Water) (Figure 2b). Sea ice import from the Nansen Basin and Kara Sea is the most important source of freshwater in the northern Barents Sea (Lind et al., 2016;Ellingsen et al., 2009). The Barents Sea is key mixing region of AW and ArWs (Porter et al., 2020). The transition between these water masses is marked by the Polar Front, which can be identified from the sea surface temperature gradient (Barton et al., 2018;Oziel et al., 2016) (Figure 1b).

Intense cooling of AW across the Barents Sea, reinforced by brine rejection due to ice formation creates dense Barents Sea Water (BSW) that cascades into the deeper troughs of the central and eastern Barents Sea (Arthun et al., 2011;Oziel et al., 2016). BSW eventually leaves the shelf, mainly through St. Anna Trough (Smedsrud et al., 2013), where it is entrained into Arctic Intermediate Water and spreads further into the Arctic basin (Schauer et al., 1997).

The Barents Sea is experiencing a rapid decline in winter and summer sea ice cover (Onarheim and Arthun, 2017;Arthun et al., 2012), full-depth warming driven by both increased ocean heat transport from the Atlantic and amplified atmospheric warming over the Arctic (Arthun et al., 2012;Onarheim et al., 2015;Serreze et al., 2009), alongside increases in salinity (Lind et al., 2018;Barton et al., 2018). The area occupied by AW is increasing and the southern expression of the Polar Front is moving north (Oziel et al., 2016;Oziel et al., 2020). In the northern Barents Sea, a reduction in sea-ice import and therefore a loss of freshwater is weakening stratification and enhancing vertical mixing (Lind et al., 2018). The northern Barents Sea is therefore transitioning from a cold, salinity stratified shelf into a warmer, less stratified Atlantic dominated climate regime (Lind et al., 2018), a process described as 'Atlantification'. These changes may increase nutrient availability to phytoplankton over the growing season (Henley et al., 2020; Randelhoff et al., 2018), which is increasingly a control on Arctic net primary production (NPP) (Lewis et al., 2020).

On the other side of the Arctic, the Pacific Ocean supplies high concentrations of nutrients onto the Chukchi and East Siberian shelves, fuelling productivity and nutrient uptake (Granger et al., 2011). Increases in volume transport through the Bering Strait in recent years (Woodgate, 2018) have increased Pacific nutrient supply to the Arctic basin. These waters are relatively deplete in nitrate (in comparison to phosphate), and combined with sedimentary denitrification on the shallow shelves (Fripiat et al., 2018;Granger et al., 2018), promote nitrogen limitation in the western Arctic Ocean (Mills et al., 2018).

Although many studies have found the western Arctic Ocean to be strongly N limited (Mills et al., 2018;Granger et al., 2018;Brown et al., 2015), we know less about the extent of N limitation and occurrence of sedimentary denitrification in the eastern Arctic Ocean. Nitrate isotope measurements can give integrated estimates of nitrogen cycling processes, yet there is currently no data on the North Atlantic inputs which provide nutrients to the Arctic basin via the Barents Sea. The [15/14]N and [18/16]O in nitrate ($\delta^{15}$N-NO3 and $\delta^{18}$O-NO3, respectively) provide complementary information about nitrate uptake by phytoplankton and regeneration processes (Sigman et al., 2009b) and can be used to determine the relevant N cycling processes.

Nitrate consumption through algal uptake fractionates both N and O in a 1:1 ratio (Granger et al., 2004) with an isotope effect close to ~5‰ (Sigman et al., 2009b). Nitrogen loss through denitrification in the water column leads to enrichment in N and O isotopes in the residual nitrate pool with a fractionation of 25-30‰ (Sigman et al., 2009a). In sediments, denitrification does not usually impart a signature on nitrate isotopes as the reaction goes to completion (Sigman et al., 2003, Lehmann et al., 2007). However, multiple studies from the western Arctic and Bering Sea show that benthic denitrification can impart a

signature on the overlying water column ($\varepsilon$= 0-5‰) when high $\delta^{15}$N-NH4 is released, a process termed coupled partial nitrification-denitrification (Brown et al., 2015).


Over most of the ocean fixed nitrogen is efficiently recycled in surface waters and the regeneration of nitrate in the water column retains a $\delta^{15}$N signature from the N source (Sigman et al., 2000). The $\delta^{15}$N signature imparted on nitrate can therefore be used to identify nutrient sources from partial utilization of nutrients (Rafter et al., 2012), different water masses (Sigman et al., 2000;Tuerena et al., 2015), new N inputs (Knapp et al., 2008;Marconi et al., 2017), atmospheric inputs (Altieri et al., 2016) and rivers (Thibodeau et al., 2017).


In contrast to $\delta^{15}$N-NO3, where N atoms are internally recycled during nitrification, oxygen atoms are sourced from ambient O$_2$ and seawater, in general providing a nitrification signature of $\delta^{18}$O-H2O plus 1.1‰ (Buchwald et al., 2012;Sigman et al., 2009b). The contrasting sources of N and O atoms and thus their distinct isotopic signatures allow the relative importance of preformed and regenerated nitrate to be investigated (Rafter et al., 2013). $\delta^{18}$O-NO3 has been used to quantify the extent of regeneration on the Bering Sea shelf (Granger et al., 2011) and as evidence for the significance of nitrate regeneration in sustaining nutrient stocks on Arctic Shelves (Fripiat et al., 2018;Granger et al., 2018). The tracer $\Delta$(15-18) ($\delta^{15}$N-NO3 minus $\delta^{18}$O-NO3) captures the differences between these two isotopes, highlighting nitrate sources from different oceanic environments (Rafter et al., 2013).


When nitrate is not fully consumed in surface waters, $\delta^{15}$N-NO3 and $\delta^{18}$O-NO3 can indicate the extent of seasonal nitrate uptake by phytoplankton (DiFiore et al., 2009). The $\delta^{15}$N of surface water nitrate increases as nitrate is progressively utilized by phytoplankton, through the preferential consumption of $^{14}$N (Sigman et al., 1999). Together with other N cycling processes this can be described by Rayleigh fractionation systematics (Mariotti et al., 1981). Nitrate utilisation by phytoplankton in an environment where there is no resupply of nutrients, i.e. a stratified upper ocean in summer, follows Rayleigh fractionation systematics for a closed system, with $\delta^{15}$N-NO3 falling on a fractionation trend for its isotopic effect ($\varepsilon$) (Granger et al., 2004). In combination with dissolved nutrients, the $\delta^{15}$N of particulate nitrogen ($\delta^{15}$N-PN) can track the extent of biological utilisation, contrasting nutrient sources, and the significance of new versus regenerated nutrients (Altabet and Francois, 1994).

In this study we report the first stable isotope measurements of dissolved and particulate N in the Barents Sea and use them to understand the relative sources of nutrients fuelling contemporary Barents Sea productivity. We use stable isotope tracers to investigate how N cycling processes vary across the Barents Sea, in contrast to other Arctic shelves, and reflect upon the susceptibility of the ecosystem to climate change.

## 2 Materials and Methods

Samples were collected in the Barents Sea as part of the ARISE project (NERC Changing Arctic Ocean programme). Shipboard measurements were taken from the RRS James Clark Ross during July-August 2017 (JR16006). A 2,200 km transect was completed, comprising 59 full depth CTD casts, starting from the northern tip of Norway and ending at the shelf edge north-east of Svalbard (Figure 1a). The transect crossed the Barents Sea Opening (BSO), between Norway and the southern tip of Svalbard. Then, from Hopen Trench it continued north towards Kong Karls Land (along 30°E), and to the shelf edge and
Nansen Basin.

Standard CTD measurements and water sampling were performed using a stainless steel rosette equipped with a full sensor array and twenty-four 20-litre OTE bottles. Conductivity, temperature, and pressure were measured using a CTD system (Seabird 911+). Derived salinity was calibrated on-board with discrete samples using an Autosal 8400B salinometer
(Guildline) (Dumont et al., 2019) and an SBE43 oxygen sensor was calibrated against oxygen samples analysed using the Winkler method. A Biospherical QCP Cosine PAR sensor measured down-welling photosynthetically available radiation (PAR; 400-700 nm). We define the base of the euphotic zone to be the depth where PAR decreased to 1% of its surface value. The mean depth of the euphotic zone was 34.3 ±11.9 m.

Dissolved inorganic nutrient concentrations were determined using a Bran and Luebbe QuAAtro 5-channel auto analyser (SEAL Analytical) and AACE operating platform (V 6.1) following standard colorimetric methods with a CRM precision of 0.3%, 0.8% and 1.9% for nitrate+nitrite, phosphate and nitrite, respectively (Brand et al., 2020). Nitrate isotope samples were collected and filtered inline from the CTD using an Acropak and were frozen at -20°C until analysis. Of the 59 CTD casts in 2017, 23 were sampled for nitrate isotopes covering the full water column (Tuerena and Ganeshram, 2020). Particulate nitrogen
(PN) and $\delta^{15}$N-PN samples were collected from 21 profiles in the upper 200m, at stations where nitrate isotope samples were also collected (Norman et al., 2020). Samples were gently vacuum filtered through combusted GF/F filters (450 °C, 4hr, Whatman, 47 mm or 25 mm, nominal pore size 0.7 μm) until sufficient biomass was collected on the filter (8 to 12 L for the 47 mm diameter filters and 2 to 5 L for the 25 mm diameter filters). The filters were dried at 60 °C to remove all moisture and were stored folded and wrapped in combusted aluminium foil until return to the home laboratory where they were placed in a
-80 °C freezer until analysis.

To quantify the distinct nutrient concentrations, nutrient ratios and isotopic values of Atlantic Water (AW), Barents Sea Water (BSW) and Arctic Water (ArW), we define the water mass type of each sample using the water mass properties in Oziel et al. (2016). These are summarised in Table 1.


Additional sampling was conducted in collaboration with the Norsk Institutt for Vannforskning (NIVA, Oslo) during transits made by the general cargo vessel M/S Norbjørn between Tromsø, Norway and Longyearbyen, Svalbard. The M/S Norbjørn is a 'ship of opportunity' onto which NIVA has fitted a FerryBox system that measures a variety of parameters including, temperature and salinity at approximately 4 m depth. In addition, seawater can be collected directly from the system for further

analysis. During each 4 day transit in March, June, August and November 2018, surface seawater samples were collected for the analysis of $\delta^{15}$N of particulate organic nitrogen ($\delta^{15}$N-PN), the $\delta^{15}$N and $\delta^{18}$O of nitrate ($\delta^{15}$N-NO3, $\delta^{18}$O-NO3) and inorganic nutrients (nitrate+nitrite, nitrite, silicate, and phosphate) from 15 stations at pre-determined latitudes (Figure 1a). Seawater was filtered through combusted GF/F filters for $\delta^{15}$N-PN analysis and aliquots of the filtrate were placed into acid cleaned HDPE bottles and stored at -20 °C for the analysis of $\delta^{15}$N-NO3 and inorganic nutrients.


The isotopic composition of nitrate+nitrite ($\delta^{15}$N-NO3 and $\delta^{18}$O-NO3) was determined by the denitrifier method (Sigman et al., 2001;Casciotti et al., 2002) and following GEOTRACES protocols (Schlitzer et al., 2018). Samples were corrected using international reference standards N3 and USGS-34 (Weigand et al., 2016) and expressed in delta notation ($\delta^{15}$N-NO3 (‰ vs AIR) = ($R_{sam}/R_{std}$ -1) x 1000, $\delta^{18}$O-NO3 (‰ vs VSMOW) = ($R_{sam}/R_{std}$-1) x 1000). Standards were run in triplicate with a

reproducibility ($\sigma$) $\delta^{15}$N ±0.1‰ and $\delta^{18}$O ±0.3‰. Internal standards were analysed in each run and corrected using N3 and U34, with an inter-run standard deviation of $\delta^{15}$N ±0.1‰ and $\delta^{18}$O ±0.3‰. Nitrite concentrations in our study region ranged from 0-0.66μM, the highest concentration contributing 6% of the N+N pool. Our isotopic measurements are compared to studies where the nitrite in a sample has been removed using sulphamic acid (Granger and Sigman, 2009), to account for this, where nitrite was >2.5% of nitrate+nitrite, samples were re-run with sulphamic acid removal. For samples where nitrite was

<2.5% of nitrate+nitrite, we correct our $\delta^{18}$O-NO3 data for nitrite interference following Kemeny et al., (*2016*). $\delta^{15}$N-NO3+NO2 samples were also corrected assuming a $\delta^{15}$N-NO2 of -24‰ (Kemeny et al., 2016, Henley et al., 2017).

$\delta^{15}$N-PN was determined by EA-IRMS using a Costech Instruments Elemental Analyser coupled to Thermo Scientific Delta V Advantage mass spectrometer fitted with Conflo IV gas handling system. The instrumentation was operated using ISODAT

3.0 isotope ratio MS software. Prior to analysis the filters were wrapped in tin foil cones (OEA Laboratories) and pelletised. L-glutamic acid standards USGS 40 and USGS 41A were used as calibration standards during each analysis run. $\delta^{15}$N values obtained for USGS 40 were -4.52 ±0.08 ‰, n = 28 and for USGS 41A were 47.56 ±0.18 ‰, n= 21. A 10-point calibration using standard USGS 40 was measured to provide the linear regression equation (peak area vs expected N concentration) which was used to derive PN concentrations from the measured peak areas. μg/L concentrations were then calculated using

concentration obtained from the whole filter and volume of seawater filtered. The detection limit for PN was 10 μg.

## 3 Results

The Barents Sea Opening (BSO), between Norway and Svalbard, was dominated by saline (S > 34.8) Atlantic inflow, notably in Bear Island Trough (BIT, Figure 1a, Figure 2b). South of Spitzbergen Bank the water column was thermally stratified, with temperature exceeding 6 °C in the euphotic zone (Figure 2a). South of 72 °N, low salinity (S < 34.7) water from the Norwegian Coast Current (NCC) occupied the near surface layer (Figure 2b). The water column was fresher, colder and well mixed over the shallow Spitzbergen Bank. This marks the western-most extent of ArW and the Polar Front (Figure 1b) and coincides with strong tidal currents and topographically steered flows (Oziel et al., 2016;Sundfjord et al., 2007;Vage et al., 2014).

Dense Barents Sea Water (BSW) was observed near the seabed in Hopen Trench (Figure 2a, HT). Above it lay cooled Atlantic origin water and a thermally stratified surface layer. North of the narrow sill joining the Spitzbergen and Great Banks (Figure 2a, SB-GB), the approximate location of the Polar Front, colder (< 0 °C) and fresher (S < 37.7) ArW occupied depths below 50 m (Figure 2b). This was capped with an even fresher (S < 34) layer of sub-zero temperature melt water. This Polar Surface Layer extended southwards from the Nansen Basin, becoming progressively thinner. Below 100 m depth, over the shelf break and continental slope of the Nansen Basin, high salinity (cooled) Atlantic origin water was observed within the Boundary Current that entered the Arctic via the Fram Strait (far right of Figure 2a &b).

In the AW, nitrate concentrations were relatively homogenous below the mixed layer (11.8 ±1.8 μM) but low or below the limits of detection in the euphotic layer (Figure 2c). $NH_4^+$ concentrations were highest close to the seafloor over the Spitsbergen Bank (Figure 2d, Figure 4c). $\delta^{15}$N-NO3 and $\delta^{18}$O-NO3 were relatively homogenous in the deeper AW ($\delta^{15}$N-NO3 =5.1 ±0.1‰, $\delta^{18}$O-NO3=2.8 ±0.3‰, Figure 3a&b, Table 1) and N* was close to Redfield (N* -0.2 ±0.8 μM). As nitrate concentrations decreased into the euphotic zone, both $\delta^{15}$N-NO3 and $\delta^{18}$O-NO3 increased as a result of nitrate utilisation by phytoplankton (Figure 4 e and f).

The cooler ArW in the north of the Barents Sea had slightly lower (although not significantly different) nitrate concentrations of 10 ±1.1μM (Table 1). $NH_4^+$ concentrations were high close to the seafloor in Hopen Trench but decreased with increasing latitude. There was no significant difference in $\delta^{15}$N-NO3 or N* between AW and ArW (ArW $\delta^{15}$N-NO3 =5.1 ±0.1‰, N*= -1.0 ±0.7μM, Table 1), suggesting these nutrients also originated from the Atlantic. In contrast, $\delta^{18}$O-NO3 was 1.2‰ lower in ArW compared to AW (Figure 3b, Figure 4f).

In the BSW, $\delta^{15}$N-NO3, nitrate and N* were comparable to AW and ArW ($\delta^{15}$N-NO3 =5.1 ±0.4‰, Nitrate= 10.4 ±1.2 μM, N*=-1.1 ±1.1 μM). BSW $\delta^{18}$O-NO3 was 2.1 ±0.5‰, lower than AW but higher than ArW, reflecting a mix between these two water masses.

# 4 Discussion

## 4.1 Origin of Atlantic Water supplied to the Barents Sea

The origin of AW is important as pre-bloom nutrient concentrations, advected into the Barents Sea, set the upper limit on seasonal primary productivity. The nutrient concentration within AW is controlled by the relative contribution of nutrient-rich North Atlantic subpolar water and nutrient-poor subtropical waters that reach the Norwegian Sea together with the biological and physical transformations en-route. (Hatun et al., 2017;Rey, 2012;Johnson et al., 2013).  Here we consider the contribution of subtropical and subpolar water to the AW sampled in the Barents Sea based on known nitrate isotope end members. We

discuss the processes that the source waters are likely to have undergone and consider historical and future long-term trends in $\delta^{15}$N-NO3.

Atlantic Water sampled in the Barents Sea during this study had a nitrate concentration of 11.8 ±1.2 µM (below the mixed layer; Table 1) and a $\delta^{15}$N-NO3 of 5.1 ±0.1‰ (Table 1). This $\delta^{15}$N-NO3 is comparable to subpolar gyre thermocline nitrate of

4.8 ±0.1‰ (Peng et al., 2018), when compared over the same depth range (this study >200m 5.0 ±0.1‰). Subtropical sourced $\delta^{15}$N-NO3 (~3.9‰) is lower than subpolar $\delta^{15}$N-NO3 (Van Oostende et al., 2017) as there are significant inputs from N$_2$ fixation producing a lower $\delta^{15}$N nitrate source (Knapp et al., 2008). In comparison, isotopic measurements of subpolar nitrate and particulate nitrogen (PN) reveal the dominance of new production with local phytoplankton utilising nitrate sources from the subpolar thermocline (Peng et al., 2018;Van Oostende et al., 2017, this study), which is comparable to North Atlantic Deep

Water (NADW) (4.75-5‰) (Marconi et al., 2015).

The $\delta^{18}$O-NO3 of AW in the Barents Sea (2.8 ±0.3‰; Table 1) is high compared to NADW (1.67-2.02‰) (Marconi et al., 2015). The $\delta^{18}$O-NO3 of NADW results from regeneration leading to $\delta^{18}$O-NO3 close to the $\delta^{18}$O-H2O source plus 1.1‰ (Buchwald et al., 2012, Sigman et al., 2009b). Our characterisation of the AW nitrate that enters the Barents Sea reveals an

enrichment in $\delta^{18}$O-NO3 above a purely regenerated signal, which is also present in the subpolar gyre (Van Oostende et al., 2017). We measured a greater elevation in $\delta^{18}$O-NO3 (by 0.8 ‰) than $\delta^{15}$N-NO3 (0.1 ‰) compared to NADW. An elevation in $\delta^{18}$O-NO3 relative to deep water values from the North Atlantic demonstrates that partial nitrate assimilation followed by nitrification occurs in the subpolar North Atlantic which decreases $\delta^{15}$N-NO3 to a greater extent than $\delta^{18}$O-NO3 (Van Oostende et al., 2017;Peng et al., 2018). Our results suggest that seasonal mixing in the subpolar North Atlantic leaves an enrichment in

$\delta^{18}$O-NO3 to depths of >200m, a signal which is then transported onto the Barents Sea shelf.

In the North Atlantic, low $\delta^{15}$N-NO3 is associated with high salinity of the subtropical gyre (Knapp et al., 2008). The salinity of the AW supplied to the Barents Sea has increased in recent years (Barton et al., 2018;Oziel et al., 2016). We suggest that continued increases in salinity and the associated decrease in nitrate supply (Rey, 2012), has the potential to decrease $\delta^{15}$N-

NO3 of Arctic nitrate supply albeit to a small degree. Based upon the salinity-$\delta^{15}$N-NO3 relationship established in the wider Atlantic (Marconi et al., 2015;Schlitzer et al., 2018), the 0.05-0.1 psu change in salinity between the periods 1985-2005 and 2005-2016 (Barton et al., 2018) implies that there has been a 0.06-0.13 decrease in $\delta^{15}$N-NO3 (Pearson corr.= -0.82, df = 12, p-value = 0.0003).

## 4.2 Nitrate utilisation and limitation in the Barents Sea

In July 2017, nitrate was depleted in the euphotic zone, coinciding with an increase in both $\delta^{15}$N-NO3 and $\delta^{18}$O-NO3. The seasonal uptake of nitrate by phytoplankton fractionates $\delta^{15}$N-NO3 and $\delta^{18}$O-NO3 with an isotope effect ($\varepsilon$), close to 5‰ (Sigman et al., 2009b). This relationship can determine the relative importance of algal uptake versus other processes such as dilution and regeneration (DiFiore et al., 2006;Rafter et al., 2012). Here we find that in the Arctic, $\varepsilon$ is often muted in surface waters through dilution with nitrate-deplete freshwater.


The southern Barents Sea remains ice-free all year round and away from the Norwegian Coastal Current, the near surface salinity remains high. During the spring and summer months a warm surface mixed layer is established which triggers phytoplankton growth. As nitrate decreases, both $\delta^{15}$N-NO3 and $\delta^{18}$O-NO3 increase and algal uptake of nitrate is the dominant N cycling process occurring in the euphotic zone and is fuelled by new production (nitrate). We estimate a $\delta^{15}$N-NO3 uptake

fractionation of 4.7-4.9‰ (Figure 5a and c), with isotopic data following a trend for Rayleigh fractionation, or a closed system (Mariotti et al., 1981). This finding is anticipated since strong stratification isolated the euphotic zone from deeper waters during the time of sampling. In the northern Barents Sea, $\delta^{15}$N-NO3 and $\delta^{18}$O-NO3 increase as nitrate decreases in the euphotic zone. These waters are cooler and fresher and are likely to have undergone at least one seasonal cycle on the Barents Sea shelf, where there is evidence for nutrient regeneration (Section 4.3). We find a muted uptake fractionation in this region of 1.8‰

which is likely due to dilution of the nitrate concentration by fresh, nutrient depleted surface water (Figure 5a and c).

Increases in $\delta^{18}$O-NO3 demonstrate an uptake fractionation of ~6‰, slightly higher than estimated for $\delta^{15}$N-NO3 (Figure 5a &5b). In general, $\delta^{15}$N-NO3 and $\delta^{18}$O-NO3 increase to a similar degree at individual stations, with muted values of $\varepsilon$ in the Arctic Waters and higher values in the AWs (Figure 5). Seasonal fractionation in $\delta^{18}$O-NO3 is also slightly higher ($\varepsilon$=5.3‰)

compared to $\delta^{15}$N-NO3 ($\varepsilon$=4.2‰) (Figure 5d). Our estimates of AW uptake fractionation of ~4-8‰ for both $\delta^{15}$N-NO3 and $\delta^{18}$O-NO3 fall into the expected range for algal uptake (Tuerena et al., 2015, Sigman et al., 2009b). The higher fractionation of $\delta^{18}$O-NO3 may suggest some degree of simultaneous assimilation and nitrification co-occurring in the euphotic zone (Difiore et al., 2010).

The stable isotopic signal recorded in the Arctic marine food web is primarily dependent upon the particulate organic material produced by phytoplankton, representing the base of the food web, whose [15]N is controlled by the dissolved nutrient source.

With knowledge of the mechanism behind isotopic fractionation during nitrate uptake, and if nitrate uptake is the primary N cycling process occurring in the euphotic zone, then $\delta^{15}$N-PN may be predicted.

The JR16006 cruise was conducted during summer when the southern Barents Sea was thermally-stratified. Further north, sea-ice melt had established a fresh surface mixed layer resulting in salinity-driven stratification. Throughout the Barents Sea, particulate organic matter load was highest in the euphotic zone (average of 31.7 ±14.7 µg L$^{-1}$), and decreased to 9.5 ±3.4 µg L$^{-1}$ below 70 m (Figure 4d). We found that $\delta^{15}$N-PN in the euphotic zone in summer months, largely followed nitrate concentration, falling close to the trend for the integrated product of N uptake (Figure 4h, Figure 6a & 6b). In areas where

there was still nitrate available to phytoplankton, $\delta^{15}$N-PN was lower representing the preferential consumption of the lighter isotope. $\delta^{15}$N-PN increased to ~5‰ as the nitrate concentration approached zero, matching the AW source. Using this information, we predict how the $\delta^{15}$N-PN is likely to change in the euphotic layer following Rayleigh fractionation systematics ($\delta^{15}$N$_{mod}$) from the nutrient sources of AW and ArW.

$$\delta^{15}N_{mod} = \delta^{15}N_{initial} + \varepsilon \left( \frac{u}{(1-u)} \right) \times ln(u) \hspace{3cm} (1)$$

where u = NO3$_{observed}$ /NO3$_{initial}$, $\delta^{15}$N$_{initial}$ = 5.1‰, $\varepsilon$=4.8‰, and NO3$_{initial}$ = 11.8 µM. We find the spatial trends are captured in the modelled data with the highest modelled $\delta^{15}$N-PN where the concentrations are the lowest and vice versa (Figure 6c). Deviations from the trend, representing a lower isotopic effect, are in lower temperature samples from ArW (Figure 6c). At these locations the upper euphotic zone is salinity stratified and polar surface water dilutes the nitrate concentration. If the

ArW samples are corrected to the lower isotope effect of 1.8‰ and nitrate concentration (10 µM), as predicted from our $\delta^{15}$N-NO3 data, we find a Pearson's correlation, r=0.86, df=33, p=0 (Figure 6d).

The integrity of the relationship between particulate and dissolved species following Rayleigh uptake systematics is dependent on the environment. The different time scales represented by the isotopic composition of dissolved and particulate species,

relative degree of recycled production, and surface inputs from atmospheric deposition and N fixation, are all potential factors that can decouple this relationship (Knapp et al.,2016, Fawcett et al., 2011, Fawcett et al., 2014). Our finding that the large variability in nitrate concentration in the euphotic zone (from <0.5 to >8 µM) is captured in the $\delta^{15}$N-PN suggests that during the sampling period, nitrate was likely to be the principle N source to phytoplankton and the PN measured was largely of autotrophic origin.


These results support the finding that nitrate from the Atlantic is the primary source of nutrients to phytoplankton in surface waters and that the organic matter in the euphotic zone is principally autotrophic. When there is still nitrate readily available in surface waters the phytoplankton preferentially take up $^{14}$N and a lower $\delta^{15}$N is expressed in particulate N. As there is full

utilisation of nutrients over the growing season, we suggest that the integrated source of organic matter to the sediments and food web is ~5‰ throughout the Barents Sea.

In order to investigate any seasonal changes in the organic matter source in surface waters we consider the measurements of nitrate, PN and their isotopic ratios on the repeat transects across the BSO (Figure 6b & 7). Nitrate concentrations were highest in March, from replenishment over winter months. Nitrite remained below 0.25 μM throughout all seasons. The nitrite concentrations were lowest in March, suggesting that the intermediate products, $NH_4^+$ and $NO_2^-$ had been nitrified to nitrate (Figure 7). The highest nitrite concentrations were sampled in June and August, during or following the spring bloom, and remained high into November. $\delta^{15}$N-NO3 and $\delta^{18}$O-NO3 showed uptake driven changes in the nutrients during June and August, whereby the low nitrate concentrations coincided with heavy isotope values from the preferential consumption of the lighter isotope (Figure 5d, Figure 7d, e).

PN concentrations were low in winter and markedly increased in June and August (Figure 7c). $\delta^{15}$N-PN values from August, November and March were relatively constant at around 5‰. In June $\delta^{15}$N-PN decreased as the lighter isotope is preferentially consumed by phytoplankton (Figure 7f). The relatively constant value of 5‰ for the rest of the annual cycle reflected the AW source value of ~5‰, suggesting that there is limited new production occurring over the winter months and that $\delta^{15}$N-PN represents the integrated product of nitrate uptake from the previous growing season.

## 4.3 Nitrogen cycling processes occurring in the Barents Sea

### 4.3.1 Nitrification

As inflowing AW cools and freshens across the Barents Sea, $\delta^{18}$O-NO3 decreases from its AW source value of 2.8±0.3‰ to 1.6±0.3‰ (Figure 3b, Figure 4f). This decline is consistent with N recycling and nitrification. A range in nitrified $\delta^{18}$O nitrate values of -1.5 to 1.3‰ have been reported from nitrifier cocultures and field experiments (Buchwald et al., 2012). Previous field and modelling studies have used a nitrifying $\delta^{18}$O value of 1.1‰ plus $\delta^{18}O_{H2O}$ (Granger et al., 2013;Sigman et al., 2009b). As nitrate is regenerated, newly nitrified nitrate tracks the $\delta^{18}$O of seawater, which, in the Barents Sea is ~0.2‰ (Schlitzer et al., 2018), therefore as the proportion of regenerated nitrate increases, $\delta^{18}$O will decrease towards ~1.3‰.

The recycling of nitrate in-situ is a common feature on Arctic shelves, evidenced using nitrate isotopes on the West Siberian Shelf (Fripiat et al., 2018) and the Canadian Shelf (Granger et al., 2018). In these regions, $\delta^{18}$O-NO3 tracks $\delta^{18}$O-H2O showing the importance of N recycling in sustaining the N-limited primary production the following season. As we have characterised the Atlantic source $\delta^{18}$O-NO3, we can track the extent of nitrification across the Barents Sea to give an estimate of the proportion of regenerated nitrate on the Barents Sea shelf (Granger et al., 2013).

$$\frac{NO3_{reg}}{NO3_{tot}} = \frac{(\delta^{18}O_{meas} - \delta^{18}O_{AW})}{(\delta^{18}O_{reg} - \delta^{18}O_{AW})} \tag{2}$$

Where $\delta^{18}O_{meas}$= measured $\delta^{18}$O-NO3, $\delta^{18}O_{AW}$=2.8‰, $\delta^{18}O_{reg}$=1.3‰ and $NO3_{reg}/NO3_{tot}$= proportion of regenerated nitrate. In Figure 3d the proportion of nitrate regenerated follows the trend of $\delta^{18}$O-NO3. The proportion of nitrate regenerated remains relatively unmodified between the BSO and the Spitzbergen-Grand Banks Sill (the approximate location of the Polar Front), north of which the proportion of regenerated nitrate increases from <10% in the south, to >80% in ArW. The nutrient concentration of ArW is coupled to winter mixing, driven by atmospheric cooling and brine release during sea ice formation. The ArW experiences nutrient regeneration and nitrification over winter which works to decrease $\delta^{18}$O-NO3. As the shelf waters on the Barents Sea cool and freshen, the nitrate inventory is also replenished from the nitrification of ammonium which is supplied to the water column from sediments. The resupply and mixing of nutrients from the sediments is an important component in replenishing the N inventory. Alongside nitrification, $\Delta$(15-18) increases from ~2 to 3-4 ‰ from AW to ArW (Figure 4f). $\Delta$(15-18) captures variability between the two isotopes ($\delta^{15}$N-NO3 minus $\delta^{18}$O-NO3), and in this case an increasing $\Delta$(15-18) results from a lowering of $\delta^{18}$O-NO3 and no significant change in $\delta^{15}$N-NO3 (Table 1). These $\Delta$(15-18) values are still significantly lower than values reported from western Arctic basin and Siberian Sea where higher $\delta^{15}$N-NO3 increases $\Delta$(15-18) as a result of benthic denitrification (Fripiat et al., 2018, Granger et al., 2018).

### 4.3.2 Nitrogen resupply from sediments

Organic matter produced in surface waters will ultimately be regenerated in the water column or sink to the seafloor. The release of $NH_4^+$ from relatively shallow Arctic sediments has been noted in previous work, where the organic rich shelf sediments provide a source of $NH_4^+$ to the water column (Brown et al., 2015). Studies from the Chukchi Sea suggest there are annually varying rates of nitrification, with much higher rates in winter (Christman et al., 2011). This suggests that there is a build-up of $NH_4^+$ in summer and $NH_4^+$ concentrations decrease into winter as nitrification rates exceed $NH_4^+$ release. We found $NH_4^+$ was enhanced over the Spitzbergen Bank and in the Hopen Trough, with the highest concentrations close to the sediment rather than the euphotic zone, indicating that the sediments are releasing $NH_4^+$ to the water column.

$NH_4^+$ is generated in sediments by the ammonification of organic material and can be released by diffusive and non-diffusive fluxes (Granger et al., 2011). Previous studies have suggested that the $NH_4^+$ produced during ammonification should be similar to the organic matter source, but that there is a large isotopic effect (~14‰) associated with the nitrification of $NH_4^+$ to $NO_2^-$ (Casciotti et al., 2003). In section 4.2, we discuss the complete consumption of nitrate in the euphotic zone over a seasonal cycle. This finding suggests that over the course of the season, once all $NH_4^+$ that has been released from the sediments and

oxidised, $\delta^{15}$N-NO3 should reflect the N source (in this study: 5.1 ±0.1‰). There were a few samples with low (<4.8 ‰) $\delta^{15}$N-NO3 on the flanks of the Spitzbergen Bank, near the seabed at the head of Hopen Trench and over the SB-GB sill, which may be associated with partial N recycling processes and the retention of $^{15}$N in $NH_4^+$ (Casciotti et al., 2003). However, nitrate $\delta^{15}$N below the nitricline was relatively homogenous across our sampled transect, reflecting the AW source value of 5.1 ±0.1‰ (Table 1).

In the western Arctic, Bering Sea and East Siberian Sea the release of $NH_4^+$ from sediments leads to a decrease in $\delta^{18}$O-NO3 and an enrichment in $\delta^{15}$N-NO3 over the timescales of water mass transit (Fripiat et al., 2018;Granger et al., 2018). In these regions, remineralisation is greater than nitrification, therefore $NH_4^+$ diffuses out of the sediments which is higher in $^{15}$N, as low $\delta^{15}$N nitrified nitrate is lost to benthic denitrification in sediments (Granger et al., 2011). This process enriches $NH_4^+$ in $^{15}$N, a signature which is subsequently imparted on the overlying water column when $NH_4^+$ is released in other regions from the sediments and oxidized by nitrifiers (Brown et al., 2015). These trends are combined with concomitant decreases in N*, demonstrating the prevalence of benthic denitrification on Arctic shelves, where $\delta^{15}$N increases from ~6.5‰ at the Bering Strait (Brown et al., 2015;Lehmann et al., 2007) to 8‰ on the Canadian and Siberian Shelves (Fripiat et al., 2018;Granger et al., 2018).

If coupled partial nitrification-denitrification was occurring in the Barents Sea sediments, there should be an observed increase in $\delta^{15}$N-NO3 with the decrease in $\delta^{18}$O-NO3 through $NH_4^+$ release and nitrification. This is not evident in our dataset. (Table 1, Figure 4). Instead, we found no clear increase in $\delta^{15}$N-NO3 or decrease in N* from the AW entering the shelf, to ArW and BSW further north and east (Table 1). This finding suggests that either the process of $NH_4^+$ release from the sediments is insignificant to the water column nitrate inventory, or that in contrast to the Canadian and Siberian Shelves, the layer of low oxygen (and thus denitrification), is separated from the layer of ammonification and $NH_4^+$ release from sediments. The high $NH_4^+$, which exceeds 25% of the dissolved inorganic N inventory at the base of some profiles, suggests that $NH_4^+$ was accumulating in the water column at the time of our study.

In the Barents Sea, the shallow banks and slopes (e.g. Spitzbergen Bank) experience strong tidal and frontal currents which induce significant mixing (Sundfjord et al., 2007), and in shallower water winter convection is able to overturn the whole water column, processes that are able to remobilise and increase the oxygenation of surficial sediments. In the shallow regions we would therefore predict a deeper depth of denitrification within sediments and the faster release of $NH_4^+$ to the water column, largely by advective rather than diffusive fluxes.

The contrasting findings between this study in the Barents Sea and other Arctic shelves may result from a number of factors. The Pacific inflow supplies the much shallower Chukchi and Beaufort shelves (< 60m), with higher concentrations of

macronutrients (Granger et al., 2013). In contrast, the Atlantic inflow to the Barents Sea provides lower concentrations of macronutrients to a deeper shelf (> 100m), therefore the organic load to sediments and thus benthic denitrification is expected to be lower (Chang and Devol, 2009), implying that the nutrient inventory is proportional to production.

## 5 Summary

We show that nitrogen availability in the Barents Sea is supported through AW supply and the efficient replenishment of
nutrients through seasonal cycling processes. By the end of the growing season, all nitrate is consumed in surface waters and the $\delta^{15}$N of PN reflects the AW source. The N inventory is also dependent on the $NH_4^+$ release from sediments and nitrification. In contrast to other Arctic shelf regions, we find no evidence for benthic denitrification interacting with the water column (and no loss of N relative to P). Our results indicate that although nutrients are regenerated in the western Barents Sea, $\delta^{15}$N-NO3 does not increase, suggesting that $\delta^{15}$N-NO3 supplied to Arctic Intermediate Water may be comparable to the AW source
values. Our findings suggest $\delta^{15}$N-NO3 is unmodified in transit through the western Barents Sea. Additional samples collected in the Eastern Barents Sea and at the primary export gateway (St Anna Trough) are needed to confirm this.

Previous work has suggested that increasing NPP on Arctic shelves would increase organic matter supply to sediments and thus increase sedimentary denitrification rates (Arrigo and van Dijken, 2015). As N is the primary limiting nutrient to Arctic
phytoplankton (Mills et al., 2018), this would have downstream consequences to NPP in the central Arctic basin. Given the Barents Shelf is not currently a locale that hosts significant sedimentary denitrification and NPP here is limited by N, the future changes are likely to be different from those envisioned for other Arctic shelves. We suggest that N supply through the Barents Sea to the Arctic is likely to be determined by variability in AW inflow. Future changes in this inflow could impact the nutrient inventory transported through the Arctic Intermediate Water, impacting productivity in the central Arctic Basins where AWs
are transported.

## Author Contribution

RET wrote the manuscript, with significant input from JH, CM and RSG. RET, JH, RSG and CM designed the study. RET measured nitrate isotopes. LN measured particulate nitrogen and particulate nitrogen $\delta^{15}$N. All authors helped with fieldwork implementation and contributed to the final version of the manuscript.

**Data Availability**

Nutrient (doi:10/d8rg), nitrate isotope (doi:10/fg27) and particulate nitrogen isotope data (doi: 10/fkg8) are publicly available from the British Oceanographic Database website.

**Competing interests**

The authors declare that they have no conflict of interest.

**Funding Statement**

This work resulted from the ARISE project (NE/P006000/1 awarded to JH, NE/P006310/1 awarded to RSG and NE/P006035/1 awarded to CM), part of the Changing Arctic Ocean programme, jointly funded by the UKRI Natural Environment Research Council (NERC) and the German Federal Ministry of Education and Research (BMBF).

**Acknowledgments**

We thank Celeste Kellock for her assistance with sample collection on cruise JR16006. We also thank Patrick Rafter and an anonymous reviewer for their valuable input which has greatly improved the manuscript.

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

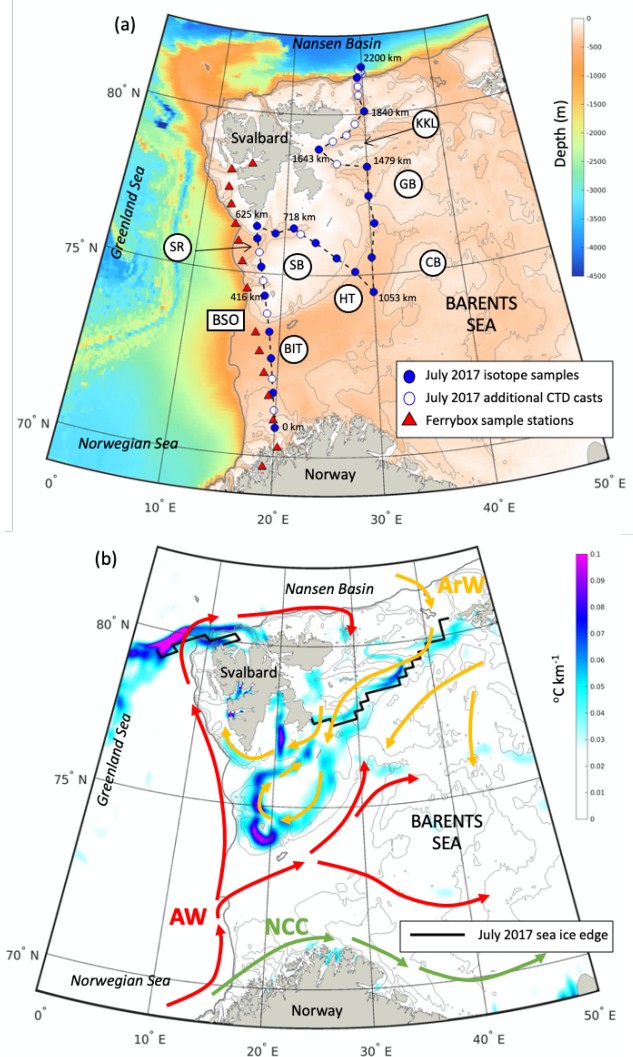

**Figure 1. (a) Station locations within the Barents Sea from JR16006 during July 2017 and on repeat FerryBox transects in March, June, August and November 2018. Shading is the depth (in metres). The grey contours mark the 200 m, 300 m and 500 m isobaths. Key bathymetric features are marked: Bear Island Trough (BIT), Hopen Trench (HT), Spitzbergen Bank (SB), Central Bank (CB), Great Bank (GB), Storfjordrenna (SR), Kong Karls Land (KKL) and the Barents Sea Opening (BSO). Distances along the transects presented in Figure 2 and 3 are marked. (b) Schematic of the circulation of Atlantic Water (AW), Arctic Water (ArW) and the Norwegian Coastal Current (NCC). Shading is the July 2017 sea surface temperature gradient (°C km⁻¹) calculated from the OSTIA SST product (Donlon et al., 2012) and shows the sea surface temperature expression of the Polar Front along the edges of Spitzbergen Bank and Grand Bank. The solid black contour marks the July 2017 sea ice edge (data from the National Snow and Ice Data Center). Grey bathymetric contours as in (a).**

**Table 1. Mean and standard deviation of nitrate concentration, N*, $\delta^{15}$N-NO3 and $\delta^{18}$O-NO3 for all samples from the base of the mixed layer and 500m classified as being either Atlantic Water, Barents Sea Water or Arctic Water according to the temperature and salinity characteristics used by Oziel et al., 2016. To account for nitrate utilisation, only samples where nitrate was within 1SD of mean nitrate for a given water mass were used to estimate $\delta^{15}$N-NO3 and $\delta^{18}$O-NO3. For reference, the salinity thresholds are contoured on Figures 2 and 3.**

| Water mass | Temp. °C | Salinity | Density kg m$^{-3}$ | Nitrate µM | N* µM | $\delta^{15}$N-NO3 ‰ | $\delta^{18}$O-NO3 ‰ |
|---|---|---|---|---|---|---|---|
| Atlantic Water (AW) (78-500m) | >3 | >34.8 | | 11.8 ±1.2 n=23 | -0.2 ±0.8 n=23 | 5.1 ±0.1 n=22 5.0±0.1 (>200m) | 2.8 ±0.3 n=20 2.6±0.3 (>200m) |
| Arctic Water (ArW) (46-500m) | <0 | <34.7 | | 10 ±1.1 n=13 | -1.0 ±0.7 n=13 | 5.1 ±0.1 n=12 | 1.6 ±0.3 n=12 |
| Barents Sea Water (BSW) (60-500m) | <2 | >34.8 | >1027.8 | 10.4 ±1.2 n=23 | -1.1 ±1.1 n=21 | 5.1±0.4 n=20 | 2.2 ±0.5 n=19 |

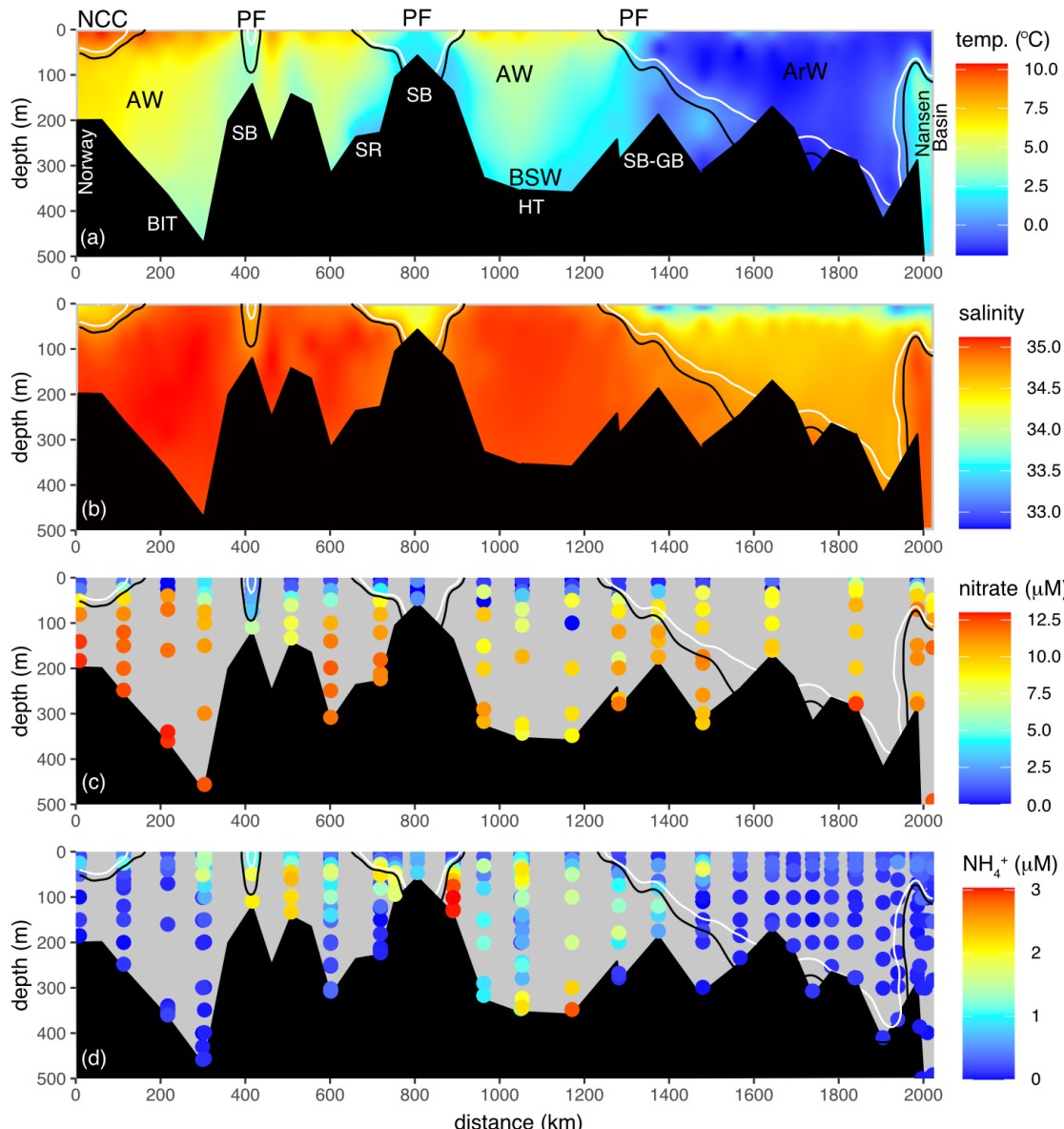

780 **Figure 2. Full 2017 transect of (a) temperature, (b) salinity, (c) nitrate, and (d) NH₄⁺. Transect is displayed in Figure 1. Solid black and white lines are the 34.8 and 34.7 isohalines. Atlantic Water (AW), Arctic Water (ArW), Barents Sea Water (BSW) and the Norwegian Coastal Current (NCC) are indicated in (a). For reference, key bathymetry features are marked in (a): Spitzbergen Bank (SB), Storfjordrenna (SR), Hopen Trench (HT), Bear Island Trough (BIT) and the Spitzbergen-Grand Bank Sill (SB-GB). PF marks the location of the surface expression of the Polar Front.**

785

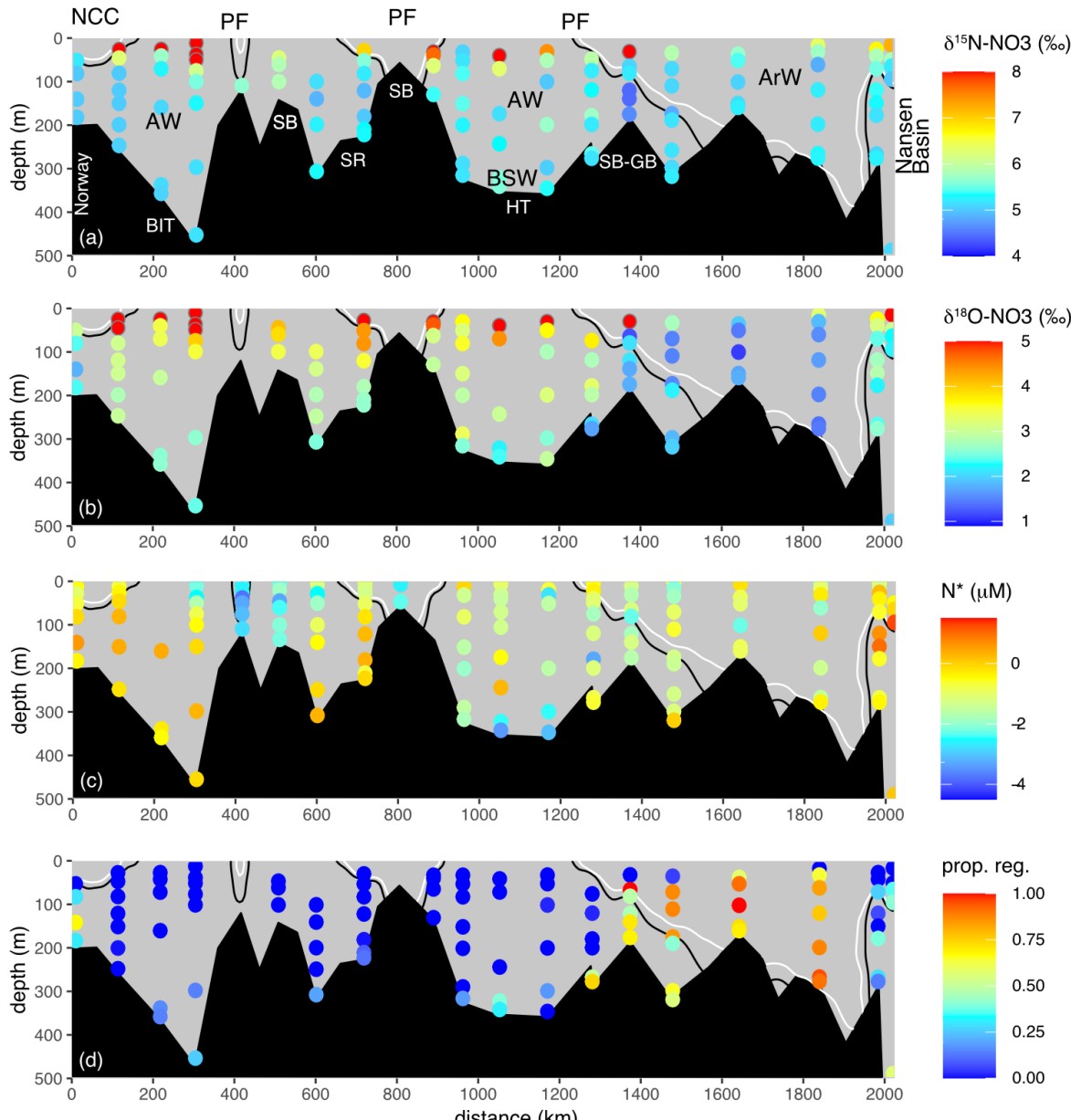

**Figure 3. Full 2017 transect of (a) $\delta^{15}$N-NO3, (b) $\delta^{18}$O-NO3, (c) N*, and (d) proportion of regenerated nitrate. Transect is displayed in Figure 1. Solid black and white lines are the 34.8 and 34.7 isohalines. Atlantic Water (AW), Arctic Water (ArW), Barents Sea Water (BSW) and the Norwegian Coastal Current (NCC) are indicated in (a). The proportion of regenerated nitrate is predicted using $\delta^{18}$O-NO3 source values from the Atlantic (2.8‰) and a nitrified value calculated using a $\delta^{18}$O-H2O of 0.2‰ (Schlitzer et al., 2018) plus 1.1‰ (1.3‰). Circles with grey outline show values not plotted to colour scale as outside of the range used in the plot.**

790

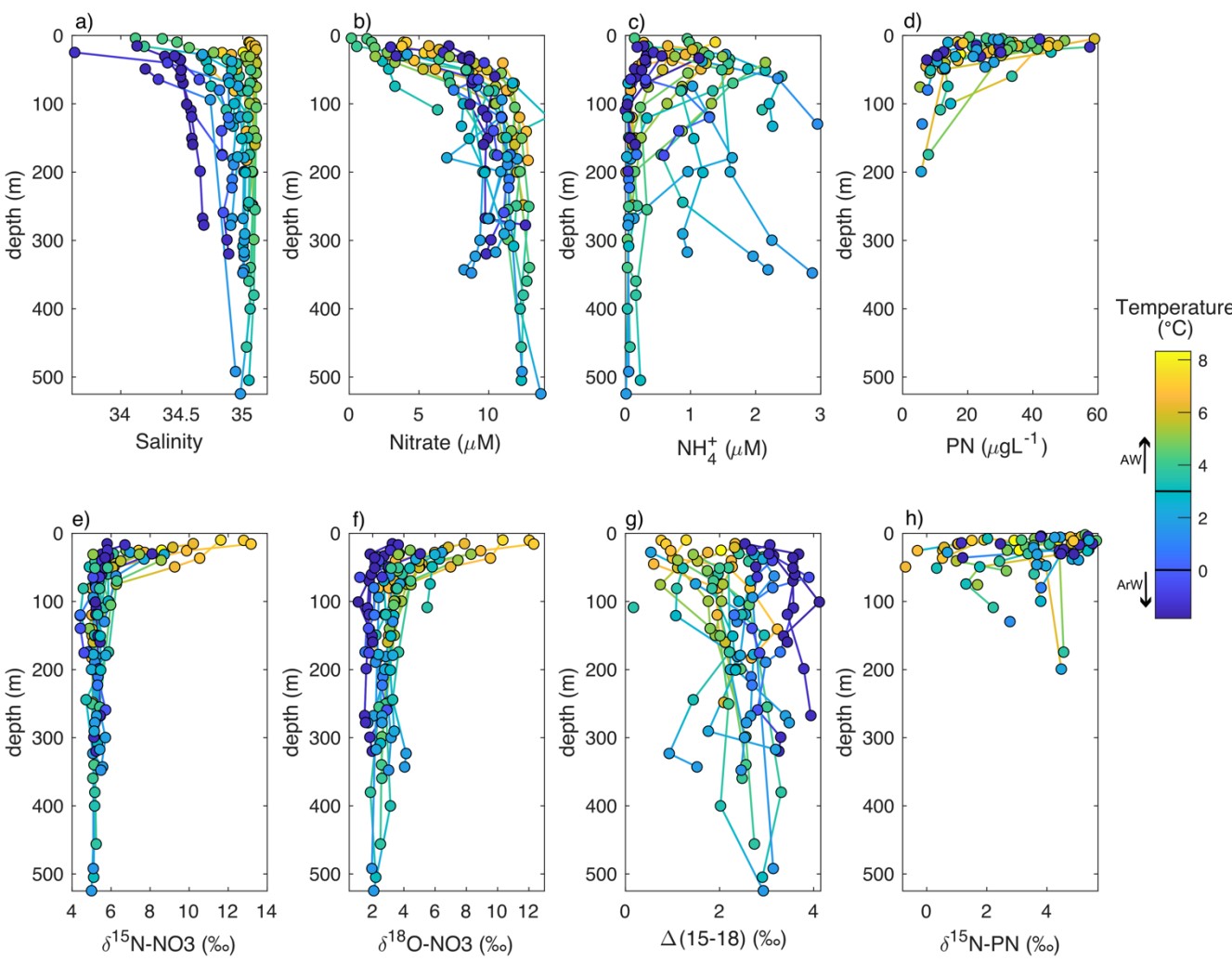

**Figure 4. Depth profiles in upper 500 m of the Barents Sea shelf. Colour denotes temperature changes. (a) Salinity, (b) Nitrate (μM), (c) NH4+ (μM), (d) PN (μg L-1), (e) δ15N-NO3 (‰ vs. AIR), (f) δ18O-NO3 (‰ vs. VSMOW), (g) Δ(15-18) (‰) and (h) δ15N-PN (‰ vs. AIR).**

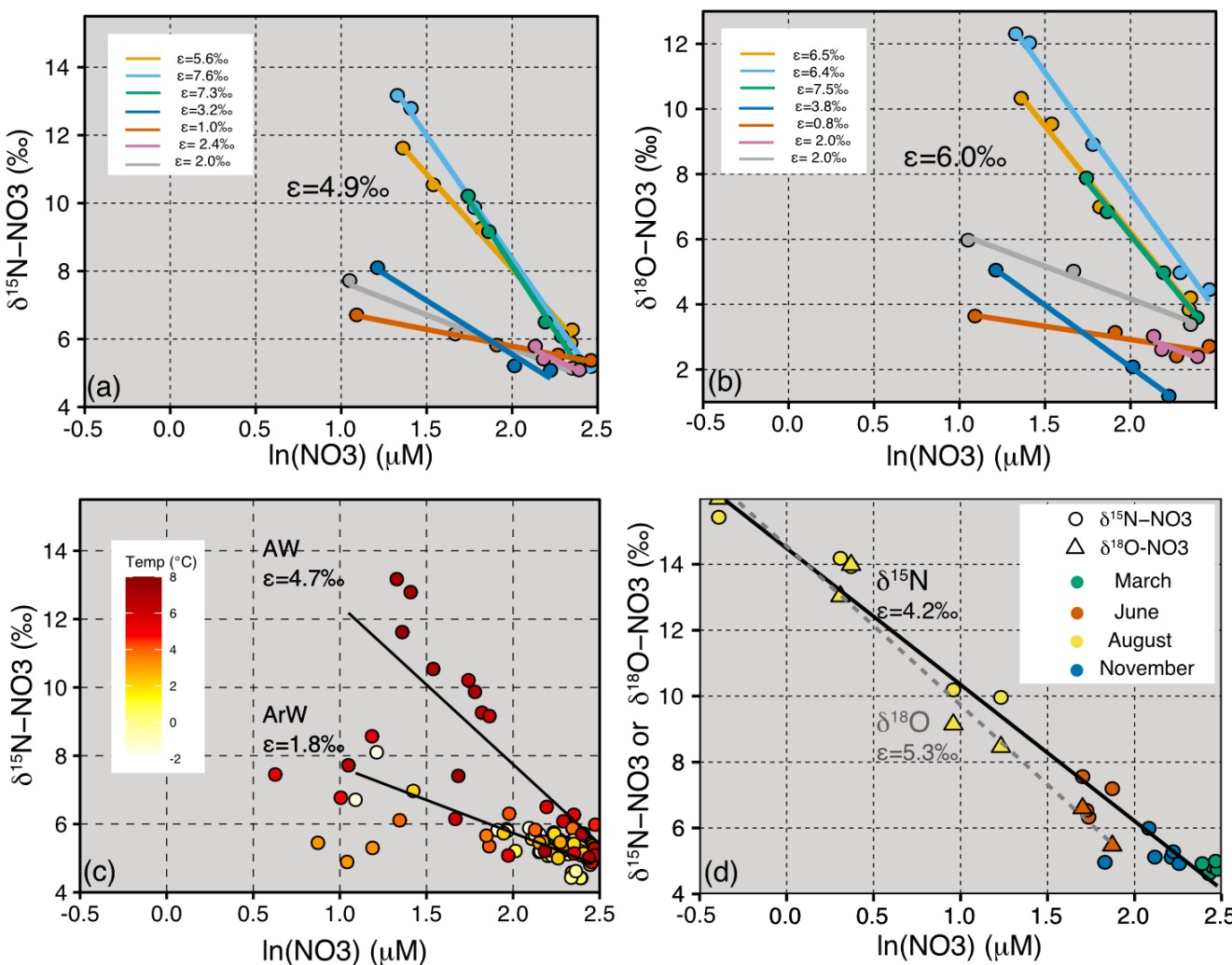

**Figure 5. (a)** $\delta^{15}$N-NO3 vs lnNO3, from JR16006 showing the isotope effect ($\varepsilon$) for individual stations and an average of 4.9‰. **(b)** $\delta^{18}$O-NO3 vs lnNO3, from JR16006 showing $\varepsilon$ for individual stations and an average of 6.0‰. In (a) and (b) only stations with 3+measurements in the upper 120m are used and $\varepsilon$ is calculated from stations with less than a 0.2 unit change in salinity. **(c)** $\delta^{15}$N-NO3 vs lnNO3 for all samples from JR16006. The two $\varepsilon$ trend lines are calculated for samples within the Atlantic Water (temperature >3°C, $\varepsilon$ = 4.7‰) and Arctic Water (temperature <0°C, $\varepsilon$ = 1.8‰). **(d)** $\delta^{15}$N-NO3 and $\delta^{18}$O-NO3 vs lnNO3 for seasonal Norbjørn samples between 72 and 76°N from the Barents Sea Opening, $\varepsilon$ =4.2‰ and 5.3‰ respectively.

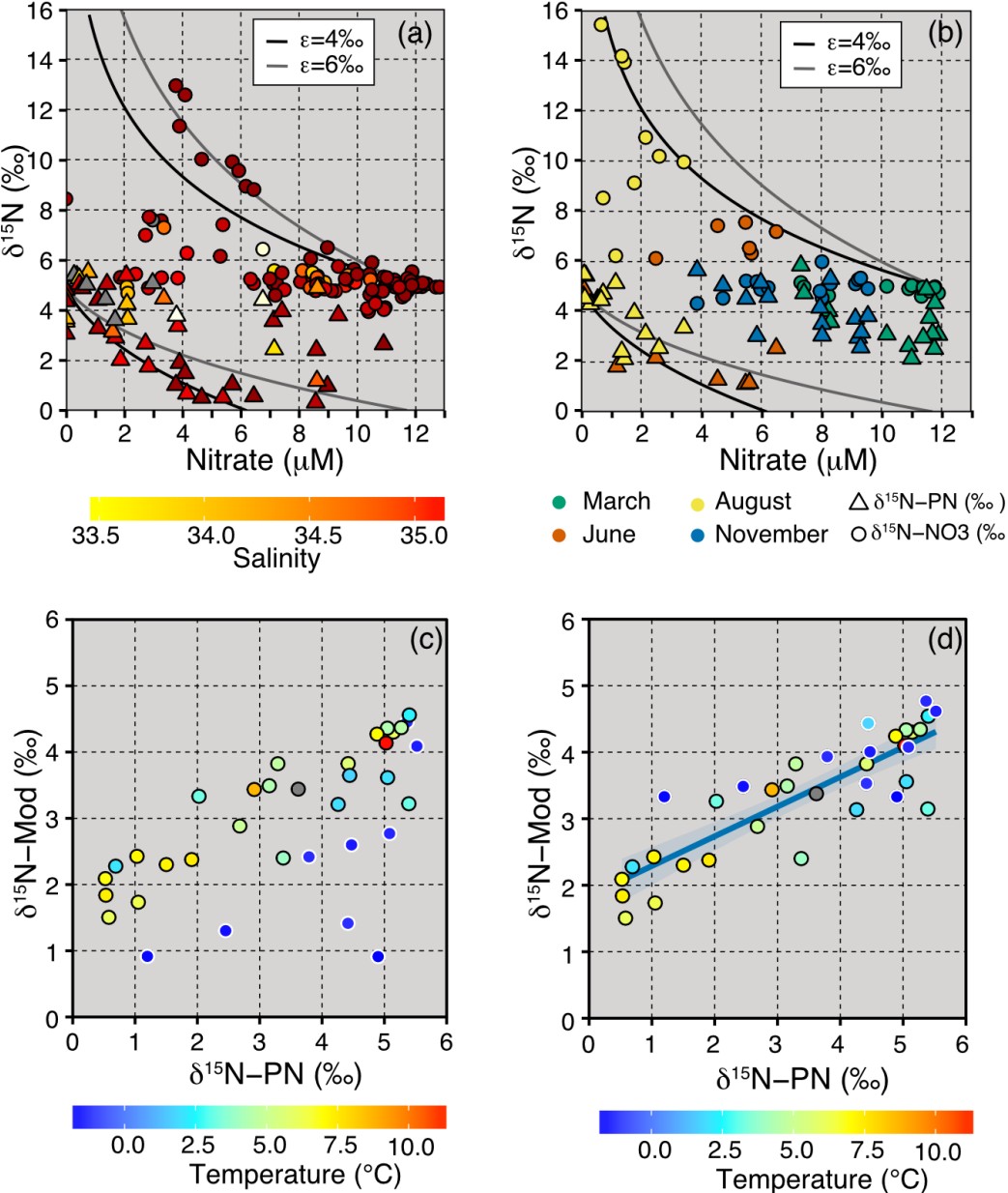

**Figure 6.** (a) $\delta^{15}$N-NO3 (circles, upper 150m) and $\delta^{15}$N-PN (triangles, upper 50m) plotted against nitrate (JR16006), with the colour axis denoting changes in salinity. The lines show the fractionation models for a closed system at $\varepsilon$=4 and 6‰ with an initial nitrate concentration of 11.8µM and $\delta^{15}$N-NO3 of 5.1‰. (b) Surface measurements of $\delta^{15}$N-NO3 (circles) and $\delta^{15}$N-PN (triangles) plotted against nitrate from the March, June, August and November Norbjørn transects, with the colour denoting the month of sampling. The lines show the fractionation models for a closed system at $\varepsilon$=4 and 6‰. (c) Regression between measured and predicted $\delta^{15}$N-PN in the upper 40m (euphotic zone) using a 4.8‰ fractionation for all samples. (d) Regression between measured and predicted $\delta^{15}$N-PN in the upper 40m (euphotic zone) using a 4.8‰ fractionation for AW and a 1.8‰ fractionation for ArW (r=0.86, df=33, p=0). ArW samples in (c) and (d) are outlined in white.

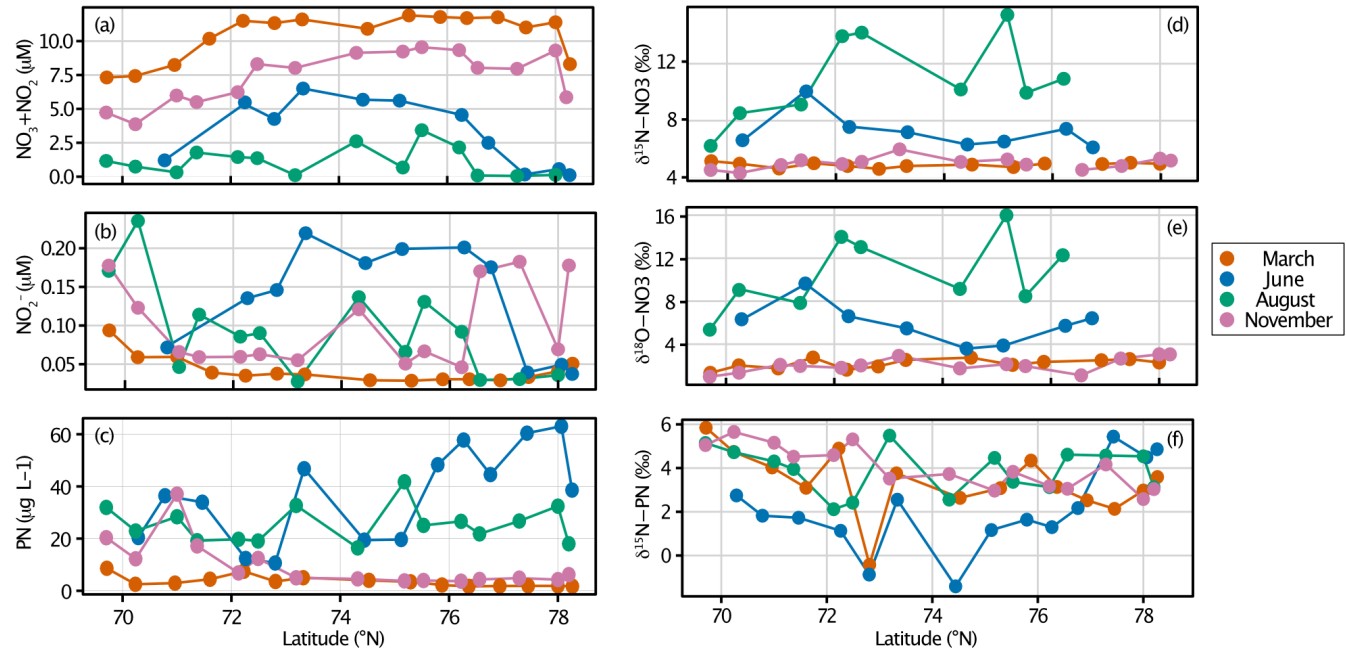

815

**Figure 7. Seasonal variability in (a) Nitrate (μM), (b) Nitrite (μM), (c) PN (μg L⁻¹), (d) δ¹⁵N-NO3(‰), (e) δ¹⁸O-NO3 (‰) and (f) δ¹⁵N-PN (‰), in surface waters along the Norbjørn Ferrybox transect of the Barents Sea Opening. The transect was completed in four different months of 2018, March = orange, June = blue, August = green, and November = pink.**