# Peer review of "Nitrate assimilation and regeneration in the Barents Sea: insights from nitrate isotopes"

_Biogeosciences, 2020_

## Referee Comment (RC1) · Patrick Rafter (Referee) · 21 Oct 2020

Review of, "Nitrate assimilation and regeneration in the Barents Sea: insights from nitrogen isotopes"

Overall, I found this to be a useful study of nitrate supply to the Barents Sea. Among the interesting results are the seasonal surface nitrate d15N across the Barents Sea Opening and the full water column nitrate isotopes and particulate matter d15N.

I would like some clarification on a couple points. First, I don't think the particulate matter sampling strategy is ever detailed—this is key to interpreting the particulate d15N measurements in relation to the nitrate isotopes. Is this entirely euphotic zone particulate d15N? Are these the same sample locations and depths as the nitrate samples? Or sub-euphotic zone? This is important—no, critical—for interpreting the particulate N measurements. I list other requests for clarification below. Given that these are necessarily suspended particulate N measurements, I found the strong relationship with nitrate d15N (and therefore connection to seasonal nitrate utilization) to be very surprising. This is because suspended N typically displays some degree of independence from surface ocean nitrate utilization (see studies by Knapp et al., Fawcett et al., and others). This question about the particulate N measurements is important. I detail other points needing clarification below.

Line by line notes

Title: It might be more accurate to describe this as insights from nitrate isotopes since it also includes nitrate O isotope measurements

Line 38-40: I understand the sentence, but others might not. Would clarify. And does the word "inflow" belong there? Line 54: Define "It" Line 68: Would insert more call-outs to figures throughout this description. Line 73: I really like this description of the water masses and their influence. Line 93: Is this acronym used more than once? 140: Wrong delta symbol? 141-142: Might want to be more explicit about why the filtration persisted until "good colour was obtained on the filter". And where exactly are these samples? Are they at the same depths as the nitrate samples? 190: Entire sentence is passive and can be clearer. Also, callout to figure (e.g., "far right in Fig. 2A & 2B) 210-216: Would begin with the "Therefore," sentence and be more active statement. 233: "in transit"? And does this refer to nitrate assimilation at depth? Or is the text calling on some earlier nitrate assimilation that was then mixed to depth? This latter seems the most likely considering seasonal North Atlantic mixing, right? There's a lot of questions here and I think this statement needs to be supported and/or described in more detail. 291: These repeat transects are very cool. 308: I would want to qualify this statement with words like, "likely represents. . ." or "is consistent with. . ." 345: This equality between nitrate d15N and sediment d15N will only be the case where there is complete nitrate consumption. I think this is the case here, but this

should be clarified. 356: released in other regions? 380: This assumes that nutrient inventory is proportional to production, right? 385: "The N inventory is also dependent" right? 393: I feel that this last part of the manuscript is pretty speculative. I think it is ok, but can see how others might have a problem with it.

―――――――――――――――――――

---

## Referee Comment (RC2) · Patrick Rafter (Referee) · 30 Oct 2020

I've read and approve of most of the authors' response to my comments. One remaining question for the authors is, how was the euphotic zone defined in the study?

---

## Short Comment (SC1) · 30 Oct 2020

Many thanks for your response.

Here I'm adding text from line 135 of the submitted version, but more information can be provided if required: 'We define the base of the euphotic zone to be the depth where PAR decreased to 1% of its surface value. The mean depth of the euphotic zone was 34.3 $\pm$11.9 m.'

kind regards, Robyn Tuerena

---

## Author Comment (AC1) · 30 Oct 2020

We thank Dr Rafter for his time in completing this review. He has contributed some very useful points to help improve the manuscript. Here we include responses to all of the comments as follows: (1) Reviewer's comment (2) Author's comment (3) Suggested change to manuscript

(1) Overall, I found this to be a useful study of nitrate supply to the Barents Sea. Among the interesting results are the seasonal surface nitrate d15N across the Barents Sea Opening and the full water column nitrate isotopes and particulate matter d15N. I would like some clarification on a couple points. First, I don't think the particulate matter sampling strategy is ever detailed. This is key to interpreting the particulate d15N measurements in relation to the nitrate isotopes. Is this entirely euphotic zone particulate d15N? Are these the same sample locations and depths as the nitrate samples? Or sub-euphotic zone? This is important, critical for interpreting the particulate N measurements. I list other requests for clarification below. Given that these are necessarily suspended particulate N measurements, I found the strong relationship with nitrate d15N (and therefore connection to seasonal nitrate utilization) to be very surprising. This is because suspended N typically displays some degree of in- dependence from surface ocean nitrate utilization (see studies by Knapp et al., Fawcett et al., and others). This question about the particulate N measurements is important. I detail other points needing clarification below.

(2) Thank you for this point, the methods section has now been edited to show exactly what data have been used in the manuscript and why, and the DOIs for the associated data (nutrients, nitrate isotopes, particulate N isotopes) will be added in the following version. Only d15N-PN samples from the euphotic zone are used in Figures 6c and d, where the modelled data is presented. In 6a, upper 50m is presented and in Figure 6b only surface data are presented (as no depth profiles were collected). To further clarify, additional comments have been added to Section 4.2 and to the figure caption for Figure 6.

The integrity of the relationship between particulate and dissolved species following Rayleigh uptake systematics is dependent on the environment. The different time scales represented by the isotopic composition of dissolved and particulate species, relative degree of recycled production, and surface inputs from atmospheric deposition and N fixation, are all potential factors that can decouple this relationship. In the oligotrophic open ocean setting (eg. Knapp et al.,2016, Fawcett et al., 2011, 2014) these processes are relatively more important.

The strong relationship in our study indicates the importance of uptake processes determining the isotopic signals. This work takes samples from the euphotic zone of a highly productive, stratified Arctic shelf, where the euphotic zone nitrate concentration

at the time of sampling, ranges from near source values (∼9uM) to near zero. In summer months on the Barents Sea shelf, if the primary nutrient source is nitrate, the d15N signature is likely to follow the sharp changes in nitrate availability and these bulk measurements are unlikely to capture variability of separate algal assemblages as seen in Fawcett et al (2011) as the primary nutrient taken up by phytoplankton at this time, is nitrate, and not ammonium.

Our Barents Sea data also shows that the nitrate-particulate d15N relationship doesn't necessarily hold up with the same trend through the study area (different trend in low temperature Arctic Waters compared to warmer Atlantic Waters), through the full season (Figure 6b), or below the euphotic zone.

(3) Added paragraph at line 285: The integrity of the relationship between particulate and dissolved species following Rayleigh uptake systematics is dependent on the environment. The different time scales represented by the isotopic composition of dissolved and particulate species, relative degree of recycled production, and surface inputs from atmospheric deposition and N fixation, are all potential factors that can decouple this relationship (Knapp et al.,2016, Fawcett et al., 2011, Fawcett et al., 2014). Our finding that the large variability in nitrate concentration in the euphotic zone (from <0.5 to >8uM) is captured in the d15N -PN suggests that during the sampling period, nitrate was likely to be the principle N source to phytoplankton and the PN measured was largely of autotrophic origin.

Line by line notes (1) Title: It might be more accurate to describe this as insights from nitrate isotopes since it also includes nitrate O isotope measurements (2) Yes we found this difficult to choose (as the paper also uses particulate N isotope measurements), but agree that the nitrate isotope measurements are the key dataset for the paper. (3) Changed to: Nitrate assimilation and regeneration in the Barents Sea: insights from nitrate isotopes

(1) Line 38-40: I understand the sentence, but others might not. Would clarify. And

does the word "inflow" belong there? (3) Suggested edit: Further insight is required into how nitrate is supplied to Arctic shelves, the nutrient cycling processes that occur in-situ and their sensitivity to climate change. A further understanding of these processes will help to inform on future changes to Arctic primary production and food web dynamics.

(1) Line 54: Define "It" Line 68: Would insert more call- outs to figures throughout this description. (3) Suggested edit: As warm and saline AW inflow water is transported across the Barents Sea, it is modified by atmospheric cooling and is mixed with cold, fresh Arctic origin water (ArW) and the Norwegian Coastal Current (NCC) (Figure 1b). ArW found across the northern Barents Sea comprises fresh Arctic river runoff, sea ice melt and precipitation, and contains the remnants of the winter mixed layer (Rudels et al., 1996). Less dense ArW isolates the sea surface and ice cover from warm AW below (Lind et al., 2016) and during the summer is capped by a well-mixed surface layer of fresh melt water (Polar Surface Water) (Figure 2b). Sea ice import from the Nansen Basin and Kara Sea is the most important source of freshwater in the northern Barents Sea (Lind et al., 2016;Ellingsen et al., 2009). The transition between AW and ArW is marked by the Polar Front, which can be identified from the sea surface temperature gradient (Barton et al., 2018;Oziel et al., 2016) (Figure 1b).

(1) Line 73: I really like this description of the water masses and their influence.

(1) Line 93: Is this acronym used more than once? (3) acronym deleted

(1) 140: Wrong delta symbol? (3) changed to small delta

(1) 141-142: Might want to be more explicit about why the filtration persisted until "good colour was obtained on the filter". And where exactly are these samples? Are they at the same depths as the nitrate samples? (2) This section has been edited to be more specific and links to DOIs will be added in the revised version. (3) Dissolved inorganic nutrient concentrations were determined using a Bran and Luebbe QuAAtro 5-channel auto analyser (SEAL Analytical) and AACE operating platform (V 6.1) following standard colorimetric methods with a CRM precision of 0.3%, 0.8% and 1.9%

for nitrate+nitrite, phosphate and nitrite, respectively (Brand et al., 2020). Nitrate isotope samples were collected and filtered inline from the CTD using an Acropak and were frozen at -20°C until analysis. Of the 59 CTD casts in 2017, 23 were sampled for nitrate isotopes covering the full water column (Tuerena and Ganeshram, 2020). d15N-PN samples were collected by gently vacuum filtering through combusted GF/F filters (450 °C, 4hr, Whatman, 48 mm or 25 mm, nominal pore size 0.7 $\mu$m) until sufficient biomass was collected on the filter (8 to 12 L for the 48 mm diameter filters and 2 to 5 L for the 25 mm diameter filters). The filters were dried at 60 °C to remove all moisture and were stored folded and wrapped in combusted aluminium foil until return to the home laboratory where they were placed in a -80 °C freezer until analysis. d15N-PN samples were collected in the upper 200m (Norman and Mahaffey, 2020), however in this study, we only utilise measurements from within the upper 50m and for exploring Rayleigh fractionation, only samples from the euphotic zone were used.

(1) 190: Entire sentence is passive and can be clearer. Also, callout to figure (e.g., "far right in Fig. 2A & 2B) (2) Sentence has been edited to read: (3) Below 100 m depth, over the shelf break and continental slope of the Nansen Basin, high salinity (cooled) Atlantic origin water was observed within the Boundary Current that entered the Arctic via the Fram Strait (far right of Figure 2a &b).

(1) 210-216: Would begin with the "Therefore," sentence and be more active statement. (2) Thank you this has definitely made this paragraph clearer, now edited to read: (3) The origin of AW is important as pre-bloom nutrient concentrations, advected into the Barents Sea, set the upper limit on seasonal primary productivity. The nutrient concentration within AW is controlled by the relative contribution of nutrient-rich North Atlantic subpolar water and nutrient-poor subtropical waters that reach the Norwegian Sea together with the biological and physical transformations en-route. (Hatun et al., 2017;Rey, 2012;Johnson et al., 2013). Here we consider the contribution of subtropical and subpolar water to the AW sampled in the Barents Sea based on known nitrate isotope end members. We discuss the processes that the source waters are likely to

have undergone and consider historical and future long-term trends in d15N-NO3.

(1) 233: "in transit"? And does this refer to nitrate assimilation at depth? Or is the text calling on some earlier nitrate assimilation that was then mixed to depth? This latter seems the most likely considering seasonal North Atlantic mixing, right? There's a lot of questions here and I think this statement needs to be supported and/or described in more detail. (2) Yes this paragraph is describing earlier nitrate assimilation occurring in the subpolar north Atlantic where there is a high degree of seasonal mixing leading to d18O enrichments to depths of >200m which are then transported onto the Barents Sea shelf. Paragraph changed to: (3) An elevation in d18O-NO3 relative to deep water values from the North Atlantic demonstrates that partial nitrate assimilation followed by nitrification occurs in the subpolar North Atlantic which decreases d15N-NO3 to a greater extent than d18O-NO3 (Van Oostende et al., 2017; Peng et al., 2018). Seasonal mixing leaves an enrichment in d18O-NO3 to depths of >200m, a signal which is then transported onto the Barents Sea shelf.

(1) 291: These repeat transects are very cool.

(1) 308: I would want to qualify this statement with words like, "likely represents. . ." or "is consistent with. . ." (3) Changed to: This decline is consistent with N recycling and nitrification.

(1) 345: This equality between nitrate d15N and sediment d15N will only be the case where there is complete nitrate consumption. I think this is the case here, but this should be clarified. (2) Yes exactly. The full utilisation of nitrate was discussed in section 4.2, specifically lines 286-289:'As there is full utilisation of nutrients over the growing season, we suggest that the integrated source of organic matter to the sediments and food web is ∼5‰ throughout the Barents Sea.' However we agree that it is worth also adding to this section. A new sentence has been added: (3) In section 4.2, we discuss the complete consumption of nitrate in the euphotic zone over a seasonal cycle. This finding suggests that over the course of the season, once all NH4+ that

has been released from the sediments and oxidised, d15N-NO3 should reflect the N source (in this study: 5.1 ±0.1‰.

(1) 356: released in other regions? (2) Sentence now reads: (3) This process enriches NH4+ in 15N, a signature which is subsequently imparted on the overlying water column when NH4+ is released in other regions from the sediments and oxidized by nitrifiers (Brown et al., 2015).

(1) 380: This assumes that nutrient inventory is proportional to production, right? (2) Yes it does, although we argue that N availability is likely the primary limiter to productivity on Arctic shelves

(1) 385: "The N inventory is also dependent" right? (2) Yes, changed to: (3)The N inventory is also dependent on the NH4+ release from sediments and nitrification.

(1) 393: I feel that this last part of the manuscript is pretty speculative. I think it is ok, but can see how others might have a problem with it. (2) Although speculative, this part puts our findings into the context of wider Arctic research about changing Arctic nutrient supply and primary production. This has particular relevance in relation to recent work regarding the importance of Arctic nutrient supply in sustaining increases in Arctic primary production (Lewis et al., 2020). We have edited the wording of this section to highlight that this is speculative. (3) Previous work has suggested that increasing NPP on Arctic shelves would increase organic matter supply to sediments and thus increase sedimentary denitrification rates (Arrigo and van Dijken, 2015). As N is the primary limiting nutrient to Arctic phytoplankton (Mills et al., 2018), this would have downstream consequences to NPP in the central Arctic basin (Lewis et al., 2020). Given the Barents Shelf is not currently a locale that hosts significant sedimentary denitrification and NPP here is limited by N, the future changes are likely to be different from those envisioned for other Arctic shelves. We suggest that N supply through the Barents sea to the Arctic is likely to be determined by variability in AW inflow. Future changes in this inflow could impact the nutrient inventory transported through the Arctic Intermediate

Water, impacting productivity in the central Arctic Basins where AWs are transported.

---

## Referee Comment (RC3) · Anonymous Referee #2 · 6 Nov 2020

In the manuscript Nitrate assimilation and regeneration in the Barents Sea: insights from nitrogen isotopes, Tuerena and colleagues present a timely study on nitrate supply and dynamics in the Barents Sea. To date the majority of work in the Arctic utilizing the isotopes of nitrate and particulate N has focused on the western Arctic Ocean, so the spatial coverage and seasonal aspects of this study will be well received by the community. Overall the manuscript is well written (and clarification is just needed in some instances), but the data seems underutilized in places and there are a number of figure panels that are never discussed, for example Figure 4f, ïĄĎ(15,18), I think this could help elevate the discussion of nitrification in this system. I have expanded on these points in detail below.

Line 20: please clarify what season you are referring to here with the phrase 'through

[Figure]

the season'.

Line 24 to 25 / 396 to 398: the foundations of this conclusion are not clear to me and hence it seems a little speculative as currently presented, please rephrase / elaborate.

Line 90: im not sure efficiently is the correct word here, I think you mean the reaction goes to completion and hence no fractionation is expressed. It would also be nice to see some of the more recent literature here that has looked at the cellular and apparent fractionation factors associated with sedimentary denitrification e.g. the work of Moritz Lehmann, Kirstin Dähnke and colleagues.

Line 140: the wrong delta has been used here.

Line 141 / 142: 48mm filters? Is this correct, or should it be 47mm?

Line 170 to 171: The correction used here needs to be clarified, what is the basis for the -24 ‰ from Kemeny et al, 2016, looking at that paper I think this value is -24 +/- 38 ‰ is this correct? Why have you only corrected the 15N values here and not the 18O, could you not assume that the 18O-NO2- would have fully exchanged with the water and use that value in a correction? I think it would also be beneficial if you could mention the nitrite concentrations observed in your samples (just the range maybe), either here or in the results (around line 191).

Line 175: Please provide information on the standards used and the reproducibility (standard deviation) of these measurements.

Line 199 to 203: here you note that there is no significant difference in 15N-NO3- or N* between AW and ArW, but note in the opening line that nitrate concentrations are lower, are nitrate concentrations significantly lower? Looking at the errors presented it doesn't look like it, please clarify and adjust language where needed (and check throughout).

Line 217: here you refer the reader to Table 1, but the values don't match and I assume that is because of the depth cut-off, please clarify and delete the references to Table 1 if needed.

Line 220 to 242: throughout this section I am a little unclear what is your contribution and what has come from the literature.

Line 228: a reference is needed here.

Line 254: why are you only discussing the 15N fractionation here, and not the 18O as well? In the introduction you take the time to introduce the idea of 1:1 relationship, so it seems surprising here that you don't take the time to talk about the 18O values shown in Figure 5b. This section would also benefit from a comparison to literature values.

Line 270: how you determined the concentrations of PON needs to be mentioned in the methods and where can the reader see this data?

Line 278: why are you not using the isotope effect that you determined in this study (I know a value of 5 is close, but it would still be nice to see you using your own value, unless there is a reason not to)?

Line 283: I think this should be Figure 6d.

Line 308 to 309: for clarity i suggest you add in an 18O to this sentence, so 'range in nitrified 18O-nitrate values'. The work of Carly Buchwald on this was not only from co-cultures but also field measurements, making this work / values even more valuable.

Section 4.3.1: how do your ïĄĎ(15,18) values fit in here (Figure 4f), it seems like a missed opportunity to not utilize this data here and also to compare it to literature values.

Line 386: communicating?

Figure 2: based on how the water masses have been characterized (Table 1 and the results text) I think the labelling is wrong in Panel A, I don't think they should all be ArW.

Figure 3 caption: it should be 'proportion of regenerated nitrate' not percentage in order to match the figure.

Figure 4: The panels need to be labeled in this figure and it seems a shame that the depth profiles of 15N-PN are not shown.

Figure 5 caption: it would be beneficial to more clearly explain what is shown in panel C.

Figure 6: For panels c and d, I think it would be beneficial to include the statistics in the figure caption In addition it could help to clearly mark the ArW points, so that the reader can clearly see the points that move between the two panels, but I understand that this might make the figure too busy, if so I suggest that the authors remind the reader in the caption that ArW is associated with the lower temperatures. Where do you discuss / utilize panel b in the text?

---

## Author Comment (AC2) · 11 Nov 2020

Authors response to review 2

We thank the reviewer for their constructive review of our manuscript. They have contributed some very useful points to help improve the manuscript. Please find all responses following on from each reviewer comment:

In the manuscript Nitrate assimilation and regeneration in the Barents Sea: insights from nitrogen isotopes, Tuerena and colleagues present a timely study on nitrate supply and dynamics in the Barents Sea. To date the majority of work in the Arctic utilizing the isotopes of nitrate and particulate N has focused on the western Arctic Ocean, so the spatial coverage and seasonal aspects of this study will be well received by the

community. Overall the manuscript is well written (and clarification is just needed in some instances), but the data seems underutilized in places and there are a number of figure panels that are never discussed, for example Figure 4f, D(15,18), I think this could help elevate the discussion of nitrification in this system. I have expanded on these points in detail below.

Line 20: please clarify what season you are referring to here with the phrase 'through the season'.

-Sentence changed to say: The ammonification of organic matter in shallow sediments resupplies N to the water column and replenishes the nitrate inventory available for the following season.

Line 24 to 25 / 396 to 398: the foundations of this conclusion are not clear to me and hence it seems a little speculative as currently presented, please rephrase / elaborate.

-These conclusions relate to that fact that our study finds uptake and recycling processes do not significantly change d15N in nitrate through the Barents Sea. Therefore, nitrate assimilation and recycling is efficient in this shelf region and sedimentary denitrification does not decrease the N inventory. This implies that the main process which could alter the nutrients supplied to Barents Sea Water and to the central Arctic basin halocine is a change in the nutrient supply within AW.

Lines 24-25, changed to read: Our results suggest that as nutrients are efficiently recycled in the Barents Sea and there is no significant loss of N through sedimentary denitrification, the changing productivity in the Barents Sea is unlikely to alter N availability on shelf or, the magnitude of N advected to the central Arctic basin. However, we suggest that the AW nutrient source ultimately determines Barents Sea productivity and changes to this supply may alter Barents Sea primary production and subsequent nutrient supply to the central Arctic Ocean.

Lines 396-398, changed to read: Given the Barents Shelf is not currently a locale that

hosts significant sedimentary denitrification and NPP here is limited by N, the future changes are likely to be different from those envisioned for other Arctic shelves. We suggest that N supply through the Barents Sea to the Arctic is likely to be determined by variability in AW inflow. Future changes in this inflow could impact the nutrient inventory transported through the Arctic Intermediate Water, impacting productivity in the central Arctic Basins where AWs are transported.

Line 90: im not sure efficiently is the correct word here, I think you mean the reaction goes to completion and hence no fractionation is expressed. It would also be nice to see some of the more recent literature here that has looked at the cellular and apparent fractionation factors associated with sedimentary denitrification e.g. the work of Moritz Lehmann, Kirstin Dähnke and colleagues.

-Changed to read: In sediments, denitrification does not usually impart a signature on nitrate isotopes as the reaction normally goes to completion and thus the isotope effect is not expressed (Sigman et al., 2003, Lehmann et al., 2007).

Line 140: the wrong delta has been used here. -changed

Line 141 / 142: 48mm filters? Is this correct, or should it be 47mm? -Changed to 47mm

Line 170 to 171: The correction used here needs to be clarified, what is the basis for the -24 ‰ from Kemeny et al, 2016, looking at that paper I think this value is -24 +/- 38 ‰ is this correct? Why have you only corrected the 15N values here and not the 18O, could you not assume that the 18O-NO2- would have fully exchanged with the water and use that value in a correction? I think it would also be beneficial if you could mention the nitrite concentrations observed in your samples (just the range maybe), either here or in the results (around line 191).

-d18O data are also corrected. Paragraph has been restructured to provide more information. The concentration range has also been added:

Nitrite concentrations in our study region ranged from 0-0.66uM, the highest concentration contributing 6% of the N+N pool. Our isotopic measurements are compared to studies where the nitrite in a sample has been removed using sulphamic acid (Granger and Sigman, 2009), to account for this, where nitrite was >2.5% of nitrate+nitrite, samples were re-run with sulphamic acid removal. For all other samples, we correct our d18O-NO3 data for the ∼25 ‰ lower d18O of N2O produced from nitrite (Kemeny et al., 2016). d15N-NO3+NO2 samples were also corrected assuming a d15N-NO2 of -24‰ (Kemeny et al., 2016, Henley et al., 2017).

Line 175: Please provide information on the standards used and the reproducibility (standard deviation) of these measurements.

-Sentence added: L-glutamic acid standards USGS 40 and USGS 41A were used as calibration standards during each analysis run. d15N values obtained for USGS 40 were -4.52 ±0.08 ‰ n = 28 and for USGS 41A were 47.56 ±0.18 ‰ n= 21.

Line 199 to 203: here you note that there is no significant difference in 15N-NO3- or N* between AW and ArW, but note in the opening line that nitrate concentrations are lower, are nitrate concentrations significantly lower? Looking at the errors presented it doesn't look like it, please clarify and adjust language where needed (and check throughout).

-The nitrate concentrations are not significantly different, the text has now been altered to address this: The cooler ArW in the north of the Barents Sea had slightly lower (although not significantly different) nitrate concentrations of 10 ±1.1uM (Table 1).

Line 217: here you refer the reader to Table 1, but the values don't match and I assume that is because of the depth cut-off, please clarify and delete the references to Table 1 if needed.

-Thank you for this, the error value has now been edited and the text edited to describe samples from below the mixed layer.

Line 220 to 242: throughout this section I am a little unclear what is your contribution and what has come from the literature.

-Our results find the same conclusions of Peng et al/Van Oostende et al, - that d18O is elevated over d15N, which is attributed to partial nitrate assimilation and nitrification. Our contribution shows that this signal is well mixed and transported onto the Barents Sea shelf, as the signature shown throughout the AW.

We have made edits to this section to make it clearer. For example: 'Our results suggest that seasonal mixing in the subpolar North Atlantic leaves an enrichment in d18O-NO3 to depths of >200m, a signal which is then transported onto the Barents Sea shelf. '

Line 228: a reference is needed here. -Added: Buchwald et al., 2012, Sigman et al., 2009b

Line 254: why are you only discussing the 15N fractionation here, and not the 18O as well? In the introduction you take the time to introduce the idea of 1:1 relationship, so it seems surprising here that you don't take the time to talk about the 18O values shown in Figure 5b. This section would also benefit from a comparison to literature values.

-This is a valid comment and we have added further discussion of the d18O fractionation from uptake: Increases in d18O-NO3 demonstrate an uptake fractionation of ∼6‰ slightly higher than estimated for d15N-NO3 (Figure 5a &5b). In general, d15N-NO3 and d18O-NO3 increase to a similar degree at individual stations, with muted values of e in the Arctic Waters and higher values in the AWs (Figure 5). Seasonal fractionation in d18O-NO3 is also slightly higher (e=5.3‰ compared to d15N-NO3 (e=4.2‰ (Figure 5d). Our estimates of AW uptake fractionation of ∼4-8‰ for both d15N-NO3 and d18O-NO3 fall into the expected range for algal uptake (Tuerena et al., 2015, Sigman et al., 2009). The higher fractionation of d18O-NO3 may suggest some degree of simultaneous assimilation and nitrification co-occurring in the euphotic zone (Difiore et al., 2010).

Line 270: how you determined the concentrations of PON needs to be mentioned in the methods and where can the reader see this data?
-The following sentences have been added to the methods section: A 10-point calibration using standard USGS 40 was measured to provide the linear regression equation (peak area vs expected N concentration) which was used to derive PN concentrations from the measured peak areas. $\mu$g/L concentrations were then calculated using concentration obtained from the whole filter and volume of seawater filtered. The detection limit for PN was 10 $\mu$g.

Line 278: why are you not using the isotope effect that you determined in this study (I know a value of 5 is close, but it would still be nice to see you using your own value, unless there is a reason not to)?

-Figure has now been changed to use an isotope effect of 4.8 per mil

Line 283: I think this should be Figure 6d.

-Thank you, yes this has now been changed

Line 308 to 309: for clarity i suggest you add in an 18O to this sentence, so 'range in nitrified 18O-nitrate values'. The work of Carly Buchwald on this was not only from co-cultures but also field measurements, making this work / values even more valuable.

-Sentence changed to now read: This decline is consistent with N recycling and nitrification. A range in nitrified d18O nitrate values of -1.5 to 1.3‰ have been reported from nitrifier cocultures and field experiments (Buchwald et al., 2012).

Section 4.3.1: how do your D(15,18) values fit in here (Figure 4f), it seems like a missed opportunity to not utilize this data here and also to compare it to literature values.

-A couple of sentences have been added here: Alongside nitrification, D(15-18) increases from ~2 to 3-4 ‰ from AW to ArW as d18O-NO3 decreases (Figure 4f). These D(15-18) values are still significantly lower than values reported from western Arctic basin and Siberian Sea highlighting that different processes are occurring in these regions (Fripiat et al., 2018, Granger et al., 2018). And further discussed in the following section.

Line 386: communicating? -Changed to interacting

Figure 2: based on how the water masses have been characterized (Table 1 and the results text) I think the labelling is wrong in Panel A, I don't think they should all be ArW.

-Thank you for this comment! Two labels have been amended to read AW.

Figure 3 caption: it should be 'proportion of regenerated nitrate' not percentage in order to match the figure. -This has been amended

Figure 4: The panels need to be labeled in this figure and it seems a shame that the depth profiles of 15N-PN are not shown.

-Labels have now been added and we have also added PN and d15NPN panels to the plot (see attached fig. 1). We have also referenced these panels in the text.

Figure 5 caption: it would be beneficial to more clearly explain what is shown in panel C. The caption has been added to read: (c) d15N-NO3 vs lnNO3 for all samples from JR16006. The two trend lines are calculated for samples within the Atlantic Water (e = 4.7‰ and Arctic Water (e = 1.8‰.

-Labels for AW and ArW also added to 5c

Figure 6: For panels c and d, I think it would be beneficial to include the statistics in the figure caption In addition it could help to clearly mark the ArW points, so that the reader can clearly see the points that move between the two panels, but I understand that this might make the figure too busy, if so I suggest that the authors remind the reader in the caption that ArW is associated with the lower temperatures.

-This is a good suggestion and we have edited the figure to distinguish the ArW points with the outline of each point highlighted in white.

Where do you discuss / utilize panel b in the text?

-Figure 6b is now referenced and discussed within section 4.2

[Figure]

[Figure]

Fig. 1.

---

## Author Comment (AC3) · 11 Nov 2020

Many thanks for your response. Here I'm adding text from line 135 of the submitted version, but more information can be provided if required: 'We define the base of the euphotic zone to be the depth where PAR decreased to 1% of its surface value. The mean depth of the euphotic zone was 34.3 ±11.9 m.' kind regards, Robyn Tuerena